# Constraints on neural redundancy

Jay A Hennig[1,2,3], Matthew D Golub[2,4], Peter J Lund[2,3,5], Patrick T Sadtler[2,6], Emily R Oby[2,6], Kristin M Quick[2,6], Stephen I Ryu[7,8], Elizabeth C Tyler-Kabara[2,9,10], Aaron P Batista[2,6†], Byron M Yu[2,4,5†], Steven M Chase[2,5†*]

[1]Program in Neural Computation, Carnegie Mellon University, Pittsburgh, United States; [2]Center for the Neural Basis of Cognition, Carnegie Mellon University, Pittsburgh, United States; [3]Machine Learning Department, Carnegie Mellon University, Pittsburgh, United States; [4]Department of Electrical and Computer Engineering, Carnegie Mellon University, Pittsburgh, United States; [5]Department of Biomedical Engineering, Carnegie Mellon University, Pittsburgh, United States; [6]Department of Bioengineering, University of Pittsburgh, Pittsburgh, United States; [7]Department of Neurosurgery, Palo Alto Medical Foundation, California, United States; [8]Department of Electrical Engineering, Stanford University, California, United States; [9]Department of Physical Medicine and Rehabilitation, University of Pittsburgh, Pittsburgh, United States; [10]Department of Neurological Surgery, University of Pittsburgh, Pittsburgh, United States

*For correspondence:
schase@cmu.edu

†These authors contributed equally to this work

Competing interests: The authors declare that no competing interests exist.

**Abstract** Millions of neurons drive the activity of hundreds of muscles, meaning many different neural population activity patterns could generate the same movement. Studies have suggested that these redundant (i.e. behaviorally equivalent) activity patterns may be beneficial for neural computation. However, it is unknown what constraints may limit the selection of different redundant activity patterns. We leveraged a brain-computer interface, allowing us to define precisely which neural activity patterns were redundant. Rhesus monkeys made cursor movements by modulating neural activity in primary motor cortex. We attempted to predict the observed distribution of redundant neural activity. Principles inspired by work on muscular redundancy did not accurately predict these distributions. Surprisingly, the distributions of redundant neural activity and task-relevant activity were coupled, which enabled accurate predictions of the distributions of redundant activity. This suggests limits on the extent to which redundancy may be exploited by the brain for computation.
DOI: https://doi.org/10.7554/eLife.36774.001

## Introduction

Neural circuits relay information from one population of neurons to another. This relay involves successive stages of downstream neurons reading out the activity of upstream neurons. In many cases, the same activity in the downstream population can be produced by different population activity patterns in the upstream population, a phenomenon termed *neural redundancy*. Redundancy is ubiquitous in neural computation, from sensory input to motor output. For example, during a task where subjects need to discriminate the color of a stimulus while ignoring its orientation (*Mante et al., 2013*), population activity patterns corresponding to the same color but different orientations are read out equivalently, and are therefore redundant. There is mounting evidence that redundancy in readouts may provide various computational benefits. For example, neural redundancy may allow us to prepare movements without executing them (*Kaufman et al., 2014*; *Elsayed et al., 2016*), enable stable computation despite unstable neural dynamics (*Driscoll et al.,*

**eLife digest** When you swing a tennis racket, muscles in your arm contract in a specific sequence. For this to happen, millions of neurons in your brain and spinal cord must fire to make those muscles contract. If you swing the racket a second time, the same muscles in your arm will contract again. But the firing pattern of the underlying neurons will probably be different. This phenomenon, in which different patterns of neural activity generate the same outcome, is called neural redundancy.

Neural redundancy allows a set of neurons to perform multiple tasks at once. For example, the same neurons may drive an arm movement while simultaneously planning the next activity. But does performing a given task constrain how often different patterns of neural activity can be produced? If so, this would limit whether other tasks could be carried out at the same time. To address this, Hennig et al. trained macaque monkeys to use a brain-computer interface (BCI). This is a device that reads out electrical brain activity and converts it into signals that can be used to control another device. The key advantage of a BCI is that the redundant activity patterns are precisely known. The monkeys learned to use their brain activity, via the BCI, to move a cursor on a computer screen in different directions.

The results revealed that monkeys could only produce a limited number of different patterns of brain activity for a given BCI cursor movement. This suggests that the ability of a group of neurons to multitask is restricted. For example, if the same set of neurons is involved in both planning and performing movements, then an animal's ability to plan a future movement will depend on the one it is currently performing.

BCIs can help patients who have suffered stroke or paralysis. They enable patients to use their brain activity to control a computer or even robotic limbs. Understanding how the brain controls BCIs will help us improve their performance and deepen our knowledge of how the brain plans and performs movements. This might include designing BCIs that allow users to multitask more effectively.

DOI: https://doi.org/10.7554/eLife.36774.002

*2017*; *Druckmann and Chklovskii, 2012*; *Murray et al., 2017*) and allow the central nervous system to filter out unwanted noise (*Moreno-Bote et al., 2014*).

To fully utilize the proposed benefits of neural redundancy, the population activity should be allowed to freely vary, as long as the readout of this activity remains consistent with task demands. This would allow the population activity to perform computations that are not reflected in the readout. However, a commonly held assumption is that neural activity might also be constrained by energetics: All things being equal, if two population activity patterns are read out equivalently, the brain should prefer the pattern that requires less energy to produce (*Laughlin et al., 1998*; *Barlow, 1969*; *Levy and Baxter, 1996*). These two lines of reasoning raise the following questions: What principles guide the production of redundant neural activity patterns? Are there constraints on which redundant activity patterns can be produced? If so, this may limit the extent to which neural circuits can exploit the proposed computational benefits of redundancy.

Redundancy has been studied extensively in motor control (*Lashley, 1933*; *Bernstein, 1967*), albeit in terms of muscular redundancy rather than neural redundancy. During arm movements, different combinations of muscle activity can lead to the same arm kinematics, meaning these different muscle activity patterns are redundant. Previous work on this muscle redundancy problem has identified two principles guiding the selection of redundant muscle activity. First, because muscle contraction requires energy in the form of ATP, the selected muscle activity should require minimum energy relative to the other redundant options (*Thoroughman and Shadmehr, 1999*; *Huang et al., 2012*; *Fagg et al., 2002*). Second, a minimal intervention strategy has been proposed in which subjects control only the aspects of muscle activity that influence the task outcome, and allow for variability in the aspects of muscle activity that do not influence the task outcome (*Scholz and Schöner, 1999*; *Todorov and Jordan, 2002*; *Valero-Cuevas et al., 2009*). To generate movements, the brain not only needs to deal with muscle redundancy, but also *neural* redundancy, which has been less studied.

One way in which neural redundancy can arise is when there are more elements (neurons or muscles) upstream than downstream. During arm movements, the activity of around thirty muscles in the arm and hand is controlled by tens of thousands of neurons in the spinal cord (*Gray, 1918*; *Feinstein et al., 1955*). Those neurons are in turn influenced by millions of neurons in the primary motor cortex and other motor areas (*Ettema et al., 1998*; *Lemon, 2008*). Thus, the neural control of arm movement is redundant (*Figure 1A*), in that different population activity patterns can generate the same movement (*Rokni et al., 2007*; *Ajemian et al., 2013*). Can the principles of muscular redundancy inform our understanding of neural redundancy?

A common challenge in studying neural redundancy is that it is typically not known which neural activity patterns are redundant, because we do not know how downstream neurons or muscles read out information. In this study we overcome this problem by leveraging a brain-computer interface (BCI), in which the activity of dozens of neurons is read out as movements of a cursor on a computer screen (*Figure 1B*) (*Taylor et al., 2002*; *Carmena et al., 2003*; *Hochberg et al., 2006*; *Ganguly and Carmena, 2009*; *Gilja et al., 2012*; *Hauschild et al., 2012*; *Sadtler et al., 2014*). A key advantage of a BCI is that the readout of the population activity (termed the BCI mapping) is fully known and defined by the experimenter (*Golub et al., 2016*). This allows us to determine precisely the redundant population activity patterns, which are those that move the cursor in exactly the same way. To illustrate this, consider a simplified example where the activity of two neurons controls a 1D cursor velocity (*Figure 1C*). The two dark green activity patterns produce the same cursor movement ($\mathbf{v}_1$), and the two light green patterns produce a different movement ($\mathbf{v}_2$). We can decompose any population activity pattern into two orthogonal components: output-potent activity and output-null activity (*Figure 1C*, black axes) (*Kaufman et al., 2014*; *Law et al., 2014*). The output-potent component determines the cursor's movement, whereas the output-null component has no effect on the cursor. Two population activity patterns are redundant if they have the same output-potent activity, but different output-null activity (e.g. the dark green square and circle on the '$\mathbf{v}_1$' dotted line in *Figure 1C*). The question we address here is, which redundant population activity patterns are preferred by the nervous system? To answer this, we assessed the distribution of output-null activity produced during each cursor movement (*Figure 1D*), and compared it to what we would expect to observe under each of several candidate hypotheses for explaining neural redundancy.

We trained three Rhesus macaques to perform a brain-computer interface task in which they controlled the velocity of a cursor on a computer screen by volitionally modulating neural activity in primary motor cortex. To understand the principles guiding the selection of redundant neural activity, we compared the observed distributions of output-null activity to those predicted by three different hypotheses. The first two hypotheses we considered were inspired by studies of muscle redundancy. First, by analogy to minimum energy principles (*Thoroughman and Shadmehr, 1999*; *Huang et al., 2012*; *Fagg et al., 2002*), neural activity may minimize unnecessary spiking (*Barlow, 1969*; *Levy and Baxter, 1996*). Second, by analogy to the minimal intervention strategy (*Scholz and Schöner, 1999*; *Todorov and Jordan, 2002*; *Valero-Cuevas et al., 2009*), output-null activity might be uncontrolled (i.e. output-potent activity is modified independently of output-null activity) because neural variability in this space has no effect on cursor movement. Third, we considered the possibility that the distribution of redundant activity may be coupled with the task-relevant activity, so that producing particular activity patterns in output-potent dimensions requires changing the distribution of activity in output-null dimensions.

We tested all hypotheses in terms of their ability to predict the distribution of output-null activity, given the output-potent activity. Hypotheses were tested within the space in which the population activity naturally resides, termed the *intrinsic manifold* (*Sadtler et al., 2014*). The results of *Sadtler et al. (2014)* indicate that neural activity cannot readily leave this manifold, and more recent results demonstrate that neural activity is further constrained by a *neural repertoire* within the intrinsic manifold (*Golub et al., 2018*). However, a repertoire defines only a set of population activity patterns, and not how often different activity patterns within the repertoire are produced. Therefore, to understand the principles governing the selection among redundant population activity patterns, we focused on predicting the *distribution* of redundant population activity within the intrinsic manifold and neural repertoire.

We found strong evidence for the third hypothesis, that redundant activity is coupled with task-relevant activity. This indicates that neural redundancy is resolved differently than muscular redundancy. Furthermore, the output-null space should not be thought of as a space in which neural

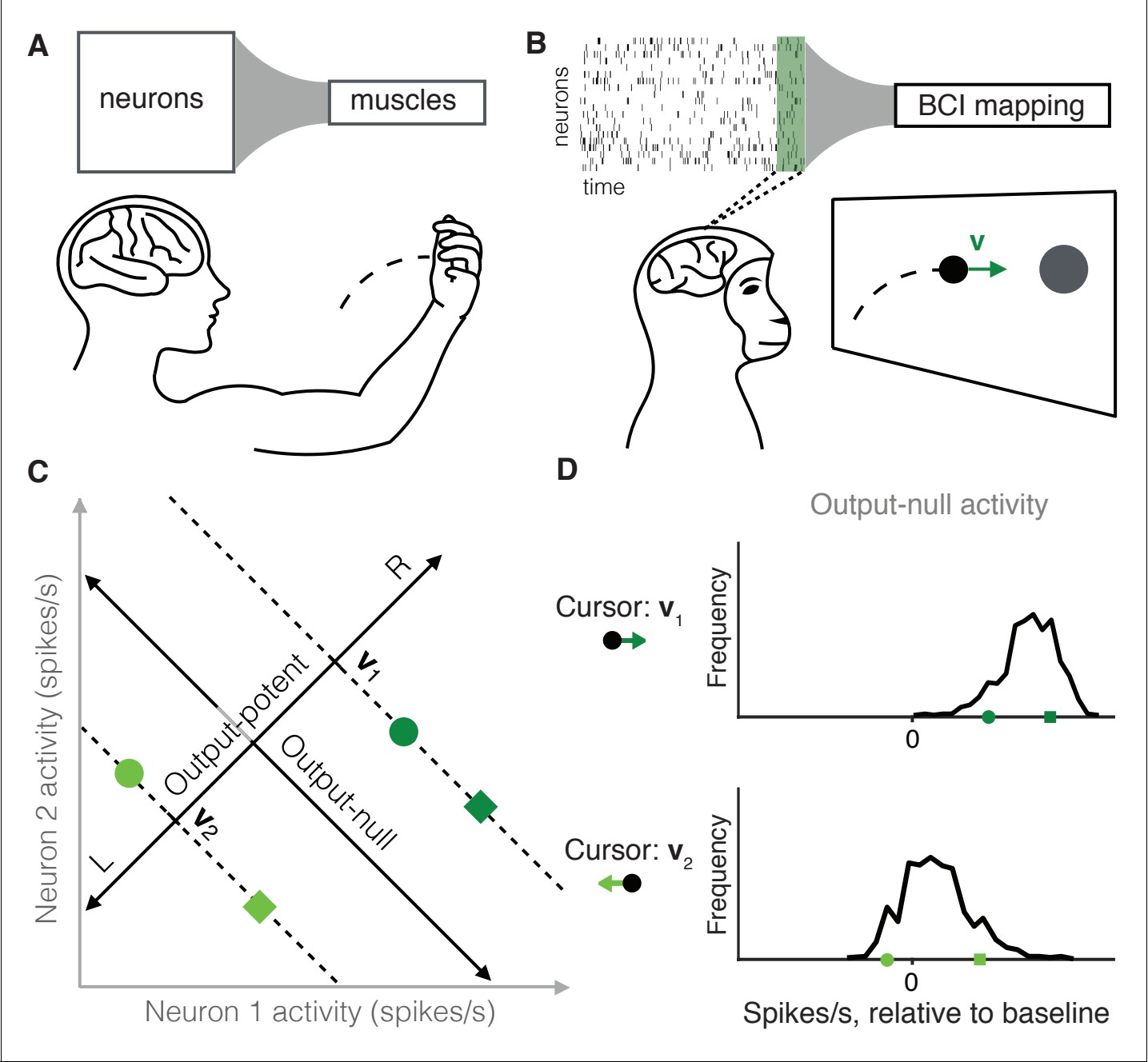

**Figure 1.** Studying the selection of redundant neural activity. (A) Millions of neurons in motor cortex drive tens of muscles to move our arms. Thus, different population activity patterns can be redundant, meaning they produce the same muscle activations and movement. (B) In a BCI, the mapping between neural activity and movement is defined by the experimenter. A subject modulates the spiking activity of tens of neurons (green rectangle) to control the 2D velocity ($\mathbf{v}$) of a cursor on a screen. (C) Example of redundant neural activity in a simplified example where the activity of two neurons (horizontal and vertical axes) drives a 1D cursor velocity (left, L, or right, R). For each of the population activity patterns shown (green squares and circles), the component of the activity along the 'Output-potent' axis determines the cursor velocity (e.g. $\mathbf{v}_1$ or $\mathbf{v}_2$), while the position of this activity along the orthogonal axis ('Output-null' axis) has no effect on the cursor's movement. Activity patterns on the same dotted line (e.g. the two dark green patterns) are redundant, because these patterns have the same output-potent activity and produce the same cursor velocity (e.g. $\mathbf{v}_1$). (D) Example distributions of neural activity along the output-null dimension (corresponding to dotted lines in (C)). Each black trace depicts the density of output-null activity observed over the course of an experiment when the cursor velocity was $\mathbf{v}_1$ (top) or $\mathbf{v}_2$ (bottom). The output-null activities of the green symbols from (C) are marked for reference. In the actual experiments, there were two output-potent dimensions and eight output-null dimensions. Output-null activity has units of spikes/s, presented relative to the vector of mean activity for each neuron ('baseline').

DOI: https://doi.org/10.7554/eLife.36774.003

*Figure 1 continued on next page*

*Figure 1 continued*

The following figure supplement is available for figure 1:

**Figure supplement 1.** Summary of behavior during the 2D center-out BCI task.

DOI: https://doi.org/10.7554/eLife.36774.004

activity can freely vary to carry out computations without regard to the output-potent activity. Instead, the distribution of output-null activity is constrained by the corresponding output-potent activity. If the required output-potent activity is defined by the task demands, this can constrain how the output-null activity can vary, and correspondingly the computations that can be carried out in the output-null space.

## Results

To study the selection of redundant neural activity, we used a BCI based on 85–94 neural units recorded using a Utah array in the primary motor cortex in each of three Rhesus macaques. Animals modulated their neural activity to move a computer cursor in a 2D center-out task (see Materials and methods; *Figure 1—figure supplement 1*). At the beginning of each experiment, we identified the 10 dimensions of the population activity that described the largest activity modulations shared among the neural units, termed the *intrinsic manifold* (*Sadtler et al., 2014*). A two-dimensional subspace of the 10-dimensional intrinsic manifold was mapped to horizontal and vertical cursor velocity and was therefore output-potent, while the eight orthogonal dimensions were output-null. Our goal was to predict the joint distribution of the observed neural activity in this eight-dimensional output-null space.

We tested several hypotheses for the selection of redundant neural activity using the following logic. First, we predicted the distributions of output-null activity expected under each hypothesis. All hypotheses' predictions were consistent with the observed behavior (i.e. the output-potent activity), and we ensured that none of these predictions required unrealistic firing rates when combined with the output-potent activity. Next, we compared the predicted distributions to the observed distributions of output-null activity to determine which hypothesis provided the best match to the observed distributions. We built the observed distributions of output-null activity as follows: At each time step during the BCI task, we assigned the recorded population activity pattern to one of eight bins corresponding to the direction of cursor movement (0°, 45°, 90°, etc.) produced by that neural activity. We binned by the cursor movement because we are studying the population activity that is redundant for a given cursor movement direction. For each bin, we projected the corresponding population activity patterns onto the eight output-null dimensions of the intrinsic manifold. The black histograms in *Figure 2*, *Figure 3*, and *Figure 4* show the marginal distributions in the first three output-null dimensions (ordered by variance accounted for). The colored histograms in *Figure 2*, *Figure 3*, and *Figure 4* are the predicted output-null distributions built under each hypothesis, which we compared to the observed distributions. The ensuing three subsections describe each hypothesis, and compare how well the corresponding predicted distributions matched the observed distributions.

During each experiment, animals controlled two different BCI mappings (i.e. the two mappings had different output-potent subspaces). The first mapping was an 'intuitive' one that required no learning for proficient control. The second mapping was a within-manifold perturbation (see Materials and methods). For the second mapping, we analyzed the trials after the behavioral performance reached asymptote. Each hypothesis predicted the distribution of output-null activity that the animal would produce under the second mapping. To form its prediction, a hypothesis could utilize the output-potent activity observed during the second mapping, as well as all neural activity recorded under control of the first mapping. This technique allowed us to avoid circularity in our results because we built the hypothesized distributions using the first behavioral context and evaluated those predictions in the second. Additionally, because animals learned to use the BCI mappings through trial and error, it is possible that the animals' assumptions about the output-null dimensions do not align perfectly with the actual output-null dimensions of the BCI mapping. To control for this, we estimated the animal's internal model of the BCI mapping (*Golub et al., 2015*). The results in the

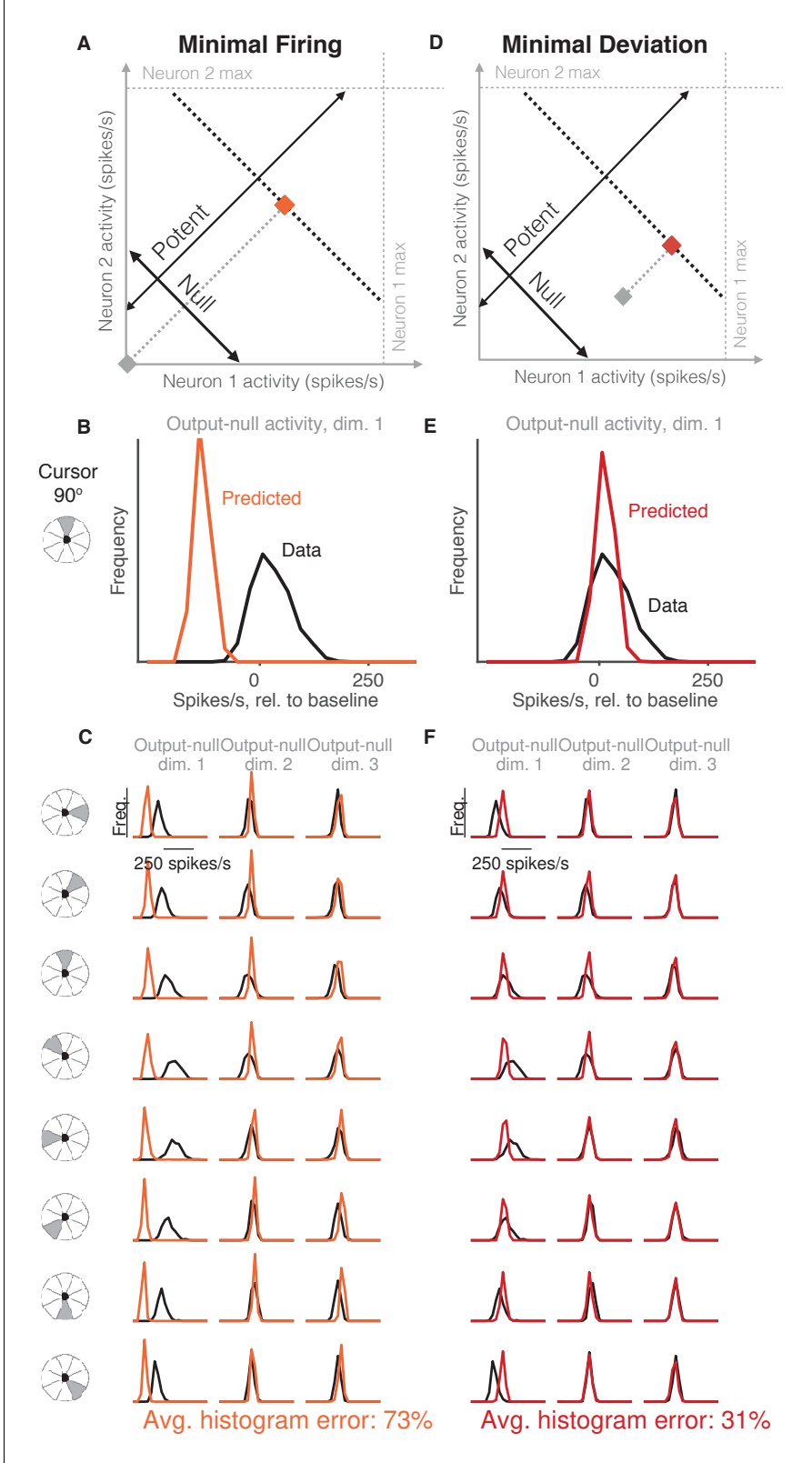

**Figure 2.** Minimal firing hypotheses. (**A**) Minimal Firing hypothesis: Given a particular output-potent activity (i.e. activity is constrained to black dotted line), subject selects the activity pattern (orange square) that requires the fewest spikes (i.e. nearest the gray square). (**B**) Distribution of observed output-null activity ('Data', in black) and activity predicted by the Minimal Firing hypothesis ('Predicted', in orange), in the first output-null dimension for upwards cursor movements. For this visualization, we applied PCA to the observed output-null activity to display the dimensions ordered by the

*Figure 2 continued on next page*

*Figure 2 continued*

amount of shared variance, with only the first of those dimensions shown here. The range of activity (e.g. ± 150 spikes/s) appears larger than that expected for a single neuron because the range tends to increase with the number of neural units contributing to that dimension. Session L20131218. (C) Distributions of observed and predicted output-null activity as in (B), for time steps when the cursor was moving in eight different directions (rows), in three (of eight) output-null dimensions explaining the most output-null variance (columns). (D) Minimal Deviation hypothesis: Given a particular output-potent activity, subject selects the activity pattern (red square) nearest a fixed population activity pattern chosen for each session by cross-validation (gray square). (E–F) Same conventions as in (B–C) for the Minimal Deviation hypothesis.

DOI: https://doi.org/10.7554/eLife.36774.005

The following source data is available for figure 2:

**Source data 1.** Histograms of predictions and data, as depicted in *Figure 2B–C* and *Figure 2E–F*.

DOI: https://doi.org/10.7554/eLife.36774.006

main text are based on this internal model, and we show in supplemental figures that all results still hold when using the actual BCI mapping.

## Minimal firing hypotheses do not accurately predict output-null activity

Previous work in motor control has found that subjects select muscle activations that minimize energy use, that is, subjects tend not to make movements with more stiffness or muscular co-contraction than necessary to complete the task (*Thoroughman and Shadmehr, 1999*; *Fagg et al., 2002*; *Huang et al., 2012*). We tested whether an analogous principle might hold true at the level of neurons (*Figure 2A*, *Minimal Firing* hypothesis). Because spiking incurs a metabolic cost (*Laughlin, 2001*; *Laughlin et al., 1998*), we first considered the hypothesis that among all the population activity patterns that produce the same cursor movement, the subject will select the one requiring the fewest spikes (*Barlow, 1969*; *Softky and Kammen, 1991*; *Levy and Baxter, 1996*).

To predict the distribution of output-null activity under this hypothesis, at each time step we found the population activity pattern that would produce the observed cursor movement with the fewest spikes across all recorded neurons (see Materials and methods). This means population activity will have minimal variability in output-null dimensions, because spiking in these dimensions does not affect cursor movement. In *Figure 2A*, the orange square depicts the activity pattern nearest zero spikes/s (gray square) among all activity patterns that would produce the same cursor movement (black dotted line). This would produce a delta distribution of output-null activity, where the delta would be located at the predicted value (orange square). To make this prediction more realistic, we incorporated Poisson spiking noise. In addition, for this hypothesis and those following, we ensured that all predictions were physiologically plausible (i.e. firing rates were between zero and the maximum rates observed in the experiment; see Materials and methods).

We constructed histograms of the output-null activity predicted by the Minimal Firing hypothesis by pooling over all time steps in which the cursor moved in a similar direction (e.g. 0°, 45°, etc.) (*Figure 2B*, orange). We compared these predicted distributions to the observed distributions of output-null activity measured for that movement direction during the experiment (*Figure 2B*, black). *Figure 2C* depicts these histograms for the same session across eight different cursor directions (rows), in three of the eight output-null dimensions (columns). For visualization, we applied principal components analysis (PCA) to display the output-null dimensions ordered by the amount of shared variance in the output-null activity. To assess how well the Minimal Firing hypothesis predicted the observed output-null activity, we computed the absolute error between the predicted and observed histograms. These errors were averaged across histograms for all eight cursor directions and eight output-null dimensions in a given session. We normalized the errors so that a perfect match between the observed and predicted histograms would result in an error of 0%, while complete mismatch between the predicted and observed histograms would yield an error of 100% (see Materials and methods). We found that the predictions of the Minimal Firing hypothesis differed from the observed activity by 73.2% ±1.3% (mean ± SE) across sessions.

One possible explanation as to why these predictions were so different from the observed activity is that minimal energy principles in the brain may not equate to minimal spiking. Perhaps a more relevant constraint is not how far the activity is away from zero firing, but rather how far the activity is from a different level of activity, such as the mean firing rate for each neuron. This alternative version

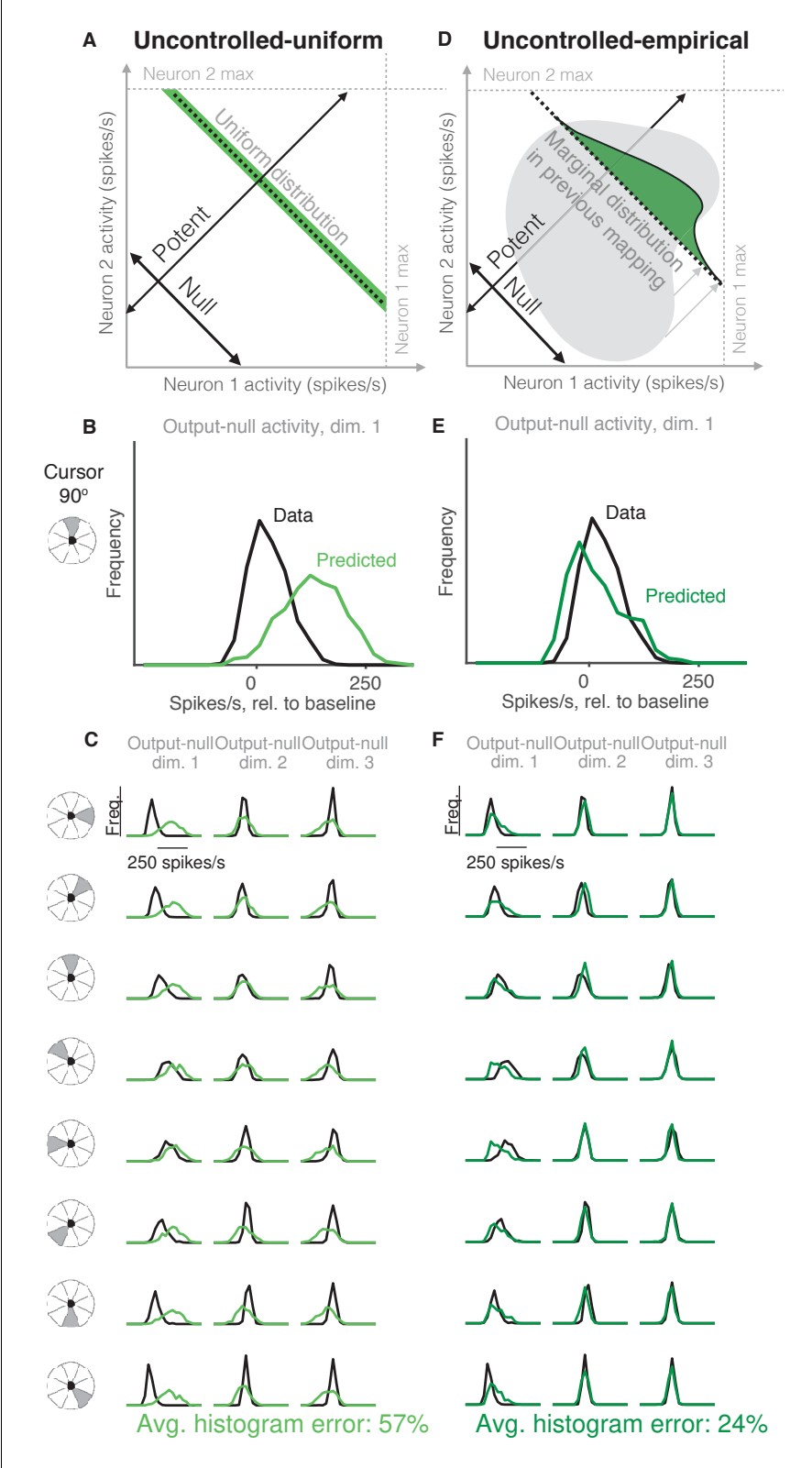

**Figure 3.** Uncontrolled hypotheses. (A) Uncontrolled-uniform hypothesis: Given a particular output-potent activity, subject selects any activity within the physiological range (dark green), sampled from a uniform distribution. (B–C) Distributions of output-null activity observed and predicted by the Uncontrolled-uniform hypothesis; same conventions as in *Figure 2*. The predicted distributions appear mound-shaped rather than uniform because we applied PCA to display the dimensions of output-null activity with the most shared variance (see Materials and methods). The range of activity increases

*Figure 3 continued on next page*

*Figure 3 continued*

with the number of neural units. Session L20131218. (**D**) Uncontrolled-empirical hypothesis: Subject selects output-null activity from the distribution of all output-null activity produced at any time while subjects used a different BCI mapping. (**E–F**) Same conventions as in (**B–C**) for the Uncontrolled-empirical hypothesis.

DOI: https://doi.org/10.7554/eLife.36774.007

The following source data is available for figure 3:

**Source data 1.** Histograms of predictions and data, as depicted in *Figure 3B–C* and *Figure 3E–F*.

DOI: https://doi.org/10.7554/eLife.36774.008

of a minimal energy hypothesis (*Figure 2D*, *Minimal Deviation* hypothesis) predicts that among all the population activity patterns that produce the same cursor movement, subjects select the one with the smallest deviation from some baseline population activity pattern. For each session, we identified the population activity pattern that would minimize the output-null prediction error across cursor directions in a cross-validated fashion (see Materials and methods) (*Figure 2E*). This hypothesis yielded an average histogram error of 30.9% ±1.2% (mean ± SE) across sessions. While this represents a substantial improvement over the Minimal Firing hypothesis (paired t-test of histogram errors in each session, $p<0.001$), the predicted distributions of output-null activity still show clear discrepancies from the observed distributions (*Figure 2F*). Thus, we sought a hypothesis that could better predict the observed distributions of output-null activity.

## Uncontrolled hypotheses do not accurately predict output-null activity

It has been shown that muscle activity exhibits more variability in output-null dimensions than in output-potent dimensions (*Scholz and Schöner, 1999*; *Todorov and Jordan, 2002*; *Valero-Cuevas et al., 2009*). An explanation of this variability asymmetry is the 'minimal intervention' principle (*Todorov and Jordan, 2002*; *Valero-Cuevas et al., 2009*; *Diedrichsen et al., 2010*), which states that while variability in output-potent dimensions should be corrected to ensure task success, variability in output-null dimensions can be left uncorrected because it does not lead to deficits in task performance. While this principle has been used to explain muscle activity, here we investigate whether it also explains neural activity. This hypothesis, that output-null activity will be 'uncontrolled' and have high variability, is in contrast to the minimal firing hypotheses, which predict that output-null activity will have low variability.

The idea that neural activity may be selected according to a minimal intervention principle does not, by itself, specify the form of the distribution in output-null dimensions. We therefore considered two specific forms of uncontrolled hypotheses. First, we supposed that if all values of output-null activity are equally likely, then output-null activity would have a uniform distribution with bounds determined by each neuron's physiological range (*Figure 3A*, *Uncontrolled-uniform*). We emphasize that the minimal intervention principle does not specify a candidate distribution, and so we consider this particular hypothesis as a limiting case, where output-null activity has maximum entropy within bounds on minimum and maximum activity. At each time step, we sampled the output-null activity from a uniform distribution within ranges observed experimentally (see Materials and methods). This procedure predicts that the output-null activity is selected independently of the current output-potent activity, reflecting the minimal intervention principle. However, note that the extent of the uniform distribution depends on the physiological range of each neuron, and so the predicted distributions of output-null activity vary slightly with the cursor direction (*Figure 3B–C*) (e.g. the length of the green bar in *Figure 3A* depends on the output-potent activity). As before, for visualization we ordered the eight output-null dimensions by the amount of shared variance explained in the recorded activity, and displayed the first three of these output-null dimensions (*Figure 3C*). Because these three dimensions were rotated along the dimensions of highest variance, the predicted histograms are mound-shaped rather than uniformly distributed (see Materials and methods). The predictions of the Uncontrolled-uniform hypothesis differed from the observed output-null activity by 56.6% ±1.1% (mean ± SE) across sessions.

In the second variant of this hypothesis, we considered a non-uniform distribution of output-null activity. If the natural variability of output-null activity is truly unmodified, then the distribution of activity observed in the same dimensions when a subject was controlling a different (previous) BCI

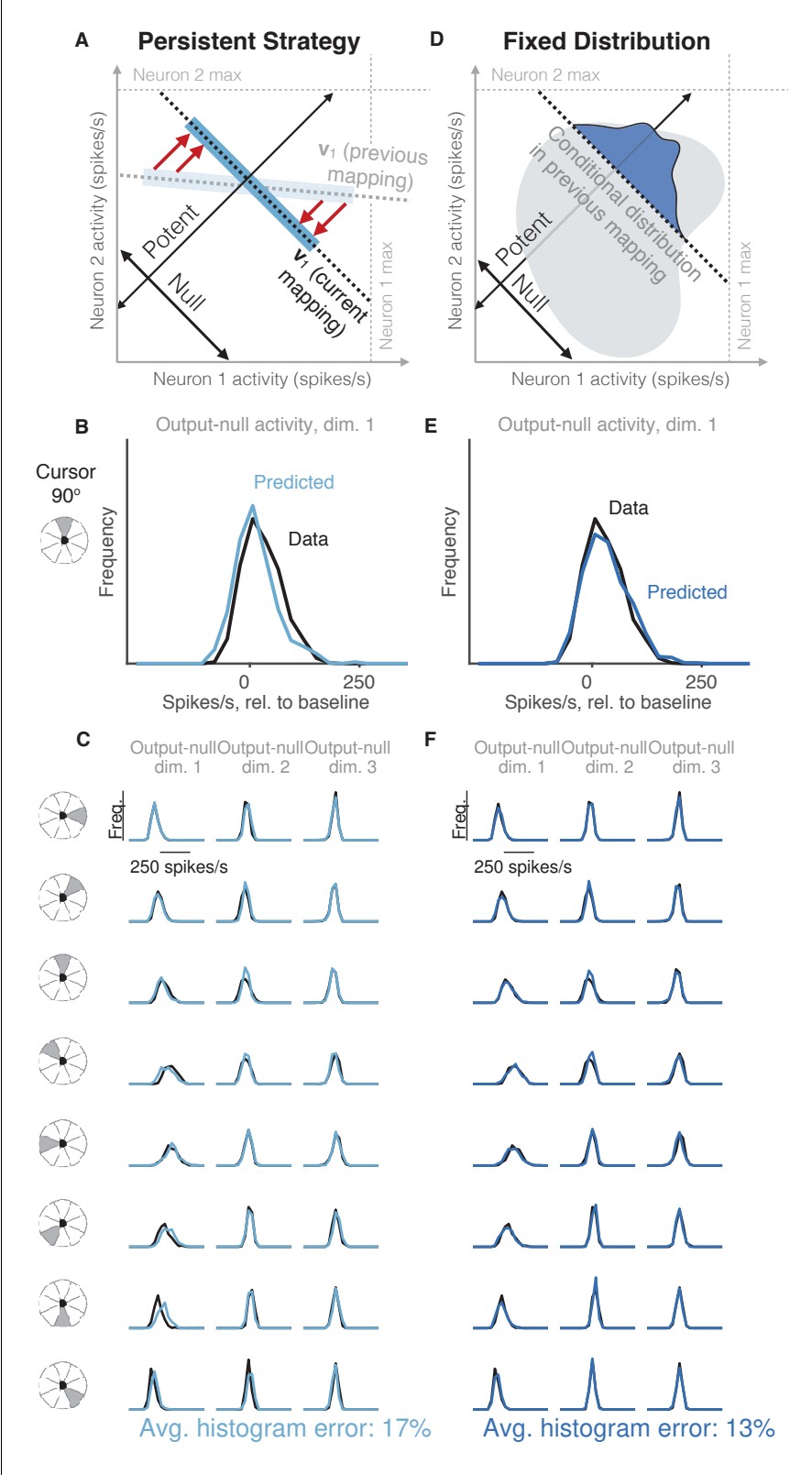

**Figure 4.** Task-transfer hypotheses. (**A**) Persistent Strategy hypothesis: Given a particular output-potent activity, subject selects an activity pattern appropriate under a different mapping (light blue rectangle), and corrects its output-potent component (red arrows) so as to produce the desired output-potent value under the current mapping (darker blue rectangle). (**B–C**) Distributions of output-null activity observed and predicted by the Persistent Strategy hypothesis; same conventions as in *Figure 2*. The range of activity increases with the number of neural units. Session L20131218. (**D**)
*Figure 4 continued on next page*

*Figure 4 continued*

Fixed Distribution hypothesis: Given a particular output-potent activity, subject selects from the output-null activity patterns that were observed concurrently with this output-potent activity while controlling a different mapping. Different patterns are selected with the same frequencies as they were under the previous mapping. (E–F) Same conventions as in (B–C) for the Fixed Distribution hypothesis.

DOI: https://doi.org/10.7554/eLife.36774.009

The following source data is available for figure 4:

**Source data 1.** Histograms of predictions and data, as depicted in *Figure 4B–C* and *Figure 4E–F*.

DOI: https://doi.org/10.7554/eLife.36774.010

mapping should have the same distribution under the current mapping (*Figure 3D*, *Uncontrolled-empirical*). Thus, under this hypothesis we construct an empirical distribution of output-null activity, which we form by projecting all of the population activity that the subject produced under the *previous* mapping onto the output-null dimensions of the *current* BCI mapping (see Materials and methods). At each time step, we sampled from this empirical distribution of output-null activity independently of the output-potent activity, again reflecting the minimal intervention principle (*Figure 3D*). We checked that combining the output-null and output-potent activity resulted in physiologically plausible population activity (see Materials and methods). If it did not, then we re-sampled a different output-null activity pattern until the combination resulted in physiologically plausible population activity. Due to this resampling, the predicted distributions of output-null activity vary slightly with the cursor direction (*Figure 3E–F*). The histograms of the predictions differed from the observed data by only 23.8% ±0.8% (mean ± SE) across sessions, which is the lowest error of all hypotheses considered so far. This suggests that previously observed population activity (in this case, recorded during use of a different BCI mapping) offers greater predictive power of the selection of output-null activity than a priori predictions such as those of the Minimal Firing, Minimal Deviation, and Uncontrolled-uniform hypotheses.

## Task-transfer hypotheses accurately predict output-null activity

Thus far, the hypothesis that best predicts the observed output-null activity is the one that uses previously observed activity to generate its predictions (Uncontrolled-empirical). This motivated us to consider more refined hypotheses that make use of this previously observed activity to generate predictions.

We first considered the hypothesis that in order to produce a desired movement, the subject selects neural activity as if he were still using the previous mapping, and corrects this activity only to ensure task success (*Figure 4A*, *Persistent Strategy*). Conceptually, when the subject wants to move the cursor in a particular direction using the current BCI mapping, he starts with the population activity patterns that he used to move the cursor in that direction under an earlier mapping (*Figure 4A*, light blue shading). Because this activity will not move the cursor in the same way that it did under the previous mapping, this activity is modified along the output-potent dimensions of the current mapping (*Figure 4A*, red arrows), reflecting the minimal intervention principle (*Todorov and Jordan, 2002*; *Valero-Cuevas et al., 2009*; *Diedrichsen et al., 2010*). This is similar to the Uncontrolled-empirical hypothesis in that we assume activity in output-null dimensions can be corrected independently of the activity in output-potent dimensions. However, instead of sampling from the entire distribution of previously observed output-null activity at each time step, here we only sample from the subset of this activity observed when subjects needed to move the cursor in the same direction as the current time step. The predictions of this hypothesis (*Figure 4B–C*) differed from the observed output-null activity by 17.4% ±0.7% (mean ± SE) across sessions.

The principle of minimal intervention posits that output-null activity can change independently from output-potent activity. Here we examine this assumption in detail. Previous work has found that the characteristic ways in which neurons covary (i.e. the dimensions of the intrinsic manifold) persist even under different BCI mappings, perhaps owing to underlying network constraints (*Sadtler et al., 2014*). All hypotheses we consider here are evaluated within the intrinsic manifold, and thus respect these constraints on population variability. Because the dimensions of the intrinsic manifold capture the variability among the neurons, it is plausible that the activity along different dimensions of the intrinsic manifold can vary independently, consistent with the minimal intervention

principle. By contrast, in the next hypothesis we consider the possibility that activity along different dimensions exhibit dependencies.

We considered the hypothesis that the distribution of activity in output-null dimensions would be predictably coupled with the activity in output-potent dimensions, even under a different BCI mapping when those dimensions were not necessarily potent and null. Under this hypothesis (*Figure 4D*, *Fixed Distribution*), given the output-potent activity, the distribution of the corresponding output-null activity remains the same as it was under a different BCI mapping (*Figure 4D*, blue frequency distribution), even if this activity was not output-null under the other mapping. This hypothesis predicts that neural activity patterns are 'yoked' across dimensions, such that producing particular activity in output-potent dimensions requires changing the distribution of activity in output-null dimensions. The histograms of output-null activity predicted by the Fixed Distribution hypothesis were a striking visual match to the recorded activity, and accurately predicted the dependence of these distributions on the cursor direction (*Figure 4E–F*). Overall, these predictions differed from the observed output-null activity by only 13.4% ±0.5% (mean ± SE) across sessions.

The Fixed Distribution hypothesis yielded a lower histogram error than all other hypotheses across sessions from three different animals (*Figure 5A*). In total, the Fixed Distribution hypothesis had the lowest histogram error in 41 of 42 sessions. The histogram error metric does not explicitly capture the degree to which hypotheses predicted the mean output-null activity, or any correlations that exist across output-null dimensions. We therefore assessed how well the predictions captured the mean and covariance of observed data in all output-null dimensions jointly (see Materials and methods). In agreement with our findings for histogram error, the mean (*Figure 5B*) and covariance (*Figure 5C*) of output-null activity was best predicted by the Fixed Distribution hypothesis, with an average mean error of 23.5 ± 1.4 spikes/s (mean ± SE) and an average covariance error of 1.4 ± 0.1 (mean ± SE in arbitrary units; see Materials and methods). These error metrics offer further evidence that the Fixed Distribution hypothesis provides a good match to the output-null distribution, as measured by the agreement between the first and second moments of the two distributions. Because these error metrics rely on a limited number of trials, they should not be compared relative to zero error. We estimated the smallest histogram, mean, and covariance errors achievable by any hypothesis, given the limited number of samples available to estimate the true output-null distributions (see Materials and methods, and gray regions in *Figure 5*). The errors of Fixed Distribution were exceedingly close to the lowest achievable error given the number of samples available (see Materials and methods). Next, we found that the Fixed Distribution hypothesis achieved the lowest prediction errors among all hypotheses when data for each monkey was considered individually (*Figure 5—figure supplement 1*). We repeated our analyses to predict output-null activity produced during the first mapping using activity observed during the second mapping (*Figure 5—figure supplement 2*). We also predicted output-null activity using the actual BCI mapping rather than the animal's internal model to define the output-null dimensions (*Figure 5—figure supplement 3*). Both analyses yielded results similar to those in *Figure 5*.

## Predicting changes in neural variability when activity becomes output-null

So far we have shown that the Fixed Distribution hypothesis provides a better explanation for the structure of output-null activity than hypotheses incorporating constraints on firing rates or the minimal intervention principle. We next sought stronger evidence for the Fixed Distribution hypothesis by assessing our predictions in the particular dimensions of population activity where it is least likely to hold. Because cursor velocity is a two-dimensional quantity, all but two dimensions of population activity for each BCI mapping are output-null. Thus, given two different BCI mappings, most dimensions will be output-null under both mappings, and so most components of the population activity have no reason to change from one mapping to the other. Therefore, we assessed whether our results held in dimensions of population activity that were output-potent during the first mapping, but output-null during the second mapping (see Materials and methods). These are the dimensions in which one would expect to see the most changes in the population activity between the first and second mappings.

Our hypotheses make distinct predictions about how the variance of activity should change if a dimension is output-potent under the first mapping and becomes output-null under the second mapping. For example, according to the Minimal Firing and Minimal Deviation hypotheses, the

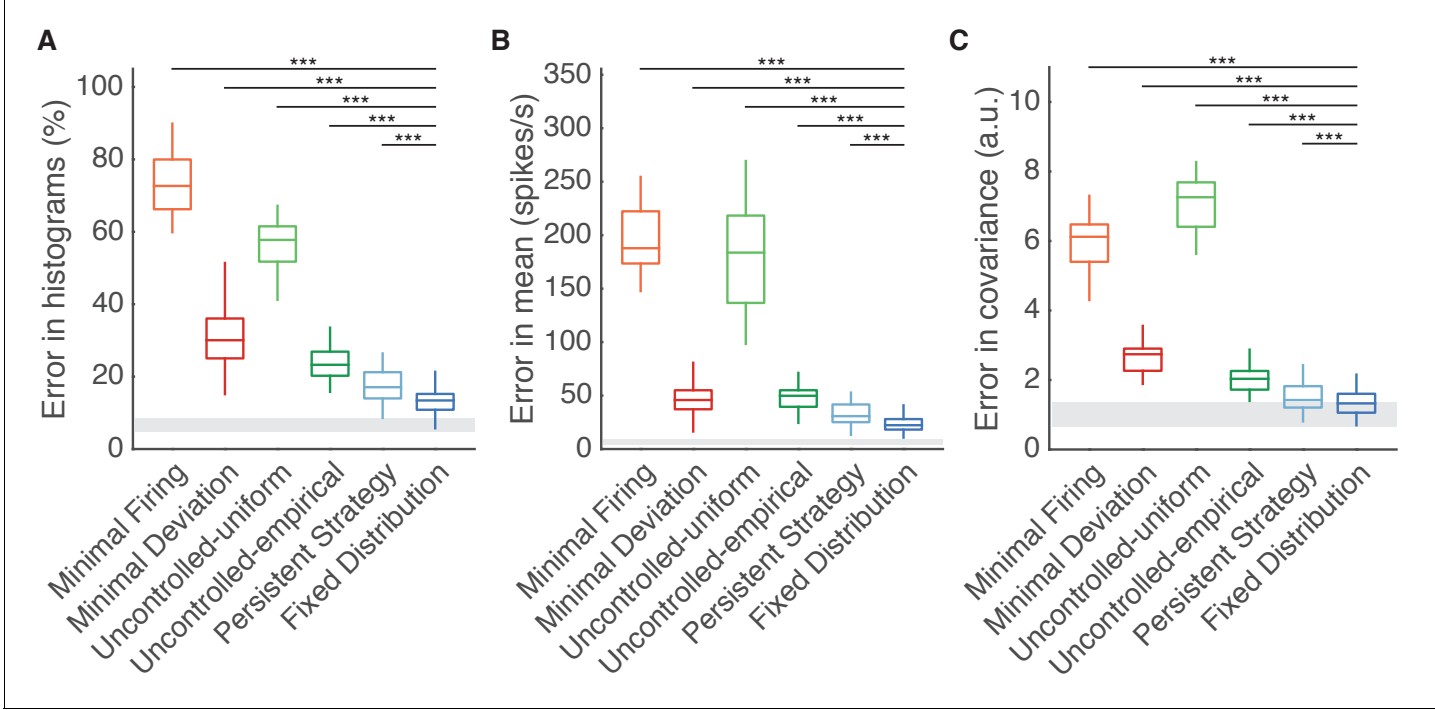

**Figure 5.** Fixed Distribution hypothesis best predicts output-null activity. Boxes depict the 25th, 50th, and 75th percentile of errors observed across sessions for all animals combined. Whiskers extend to cover approximately 99.3% of the data. Gray boxes depict the error floor across sessions (mean ± s.d.), estimated using half of the observed output-null activity to estimate the histogram, mean, and covariance of the other half (see Materials and methods). Asterisks depict a significant difference between errors of Fixed Distribution and other hypotheses for a one-sided Wilcoxon signed rank test at the $\alpha = 0.001$ (***) level. (A) Error in predicted histograms of output-null activity. For each session, histogram error was averaged across all output-null dimensions and cursor directions. Average histogram error floor was 6.7% ± 1.9% (mean ± s.d., also shown in gray). (B) Error in predicted mean of output-null activity. For each session, mean error was averaged across all cursor directions, where the mean is an 8D vector of the average activity in each output-null dimension. Average mean error floor was 6.9 ± 2.5 spikes/s (mean ± s.d., also shown in gray). (C) Error in predicted covariance of output-null activity. For each session, covariance error was averaged across all cursor directions. Average covariance error floor was 1.0 ± 0.3 (mean ± s. d., also shown in gray).

DOI: https://doi.org/10.7554/eLife.36774.011

The following source data and figure supplements are available for figure 5:

**Source data 1.** Histogram, mean, and covariance errors of all hypotheses for all sessions, as depicted in *Figure 5* and *Figure 5—figure supplement 1*.
DOI: https://doi.org/10.7554/eLife.36774.016
**Figure supplement 1.** Results for each animal.
DOI: https://doi.org/10.7554/eLife.36774.012
**Figure supplement 2.** Results when predicting output-null activity during first mapping.
DOI: https://doi.org/10.7554/eLife.36774.013
**Figure supplement 3.** Results when not using animals' internal model to define the output-null dimensions.
DOI: https://doi.org/10.7554/eLife.36774.014
**Figure supplement 4.** Identifying the animal's internal model to define the output-null dimensions.
DOI: https://doi.org/10.7554/eLife.36774.015

variance of activity will collapse in dimensions that are output-null because unnecessary spiking is undesirable. Thus, if a dimension becomes output-null, variance in this space should exhibit a marked decrease. On the other hand, the Uncontrolled hypotheses predict that, when conditioned on the cursor movement, variance will expand when the activity is output-null. This occurs because variability in this dimension will no longer affect cursor movement, and would therefore no longer need to be suppressed. Finally, the Fixed Distribution hypothesis posits that the same distributions of output-null activity will be observed regardless of whether a dimension was previously output-potent or output-null, and so this hypothesis predicts that there will be little to no change in the variance of activity in a particular dimension under the two mappings.

We asked whether the variance of population activity decreased, increased, or remained the same in dimensions that changed from being output-potent to output-null (*Figure 6A*). Critically, we computed the variance of activity after first binning by the corresponding angle in the output-potent dimensions of the *second* mapping. This was done so that the neural activity in each bin would all result in similar cursor movements under the second mapping, and is identical to the procedure

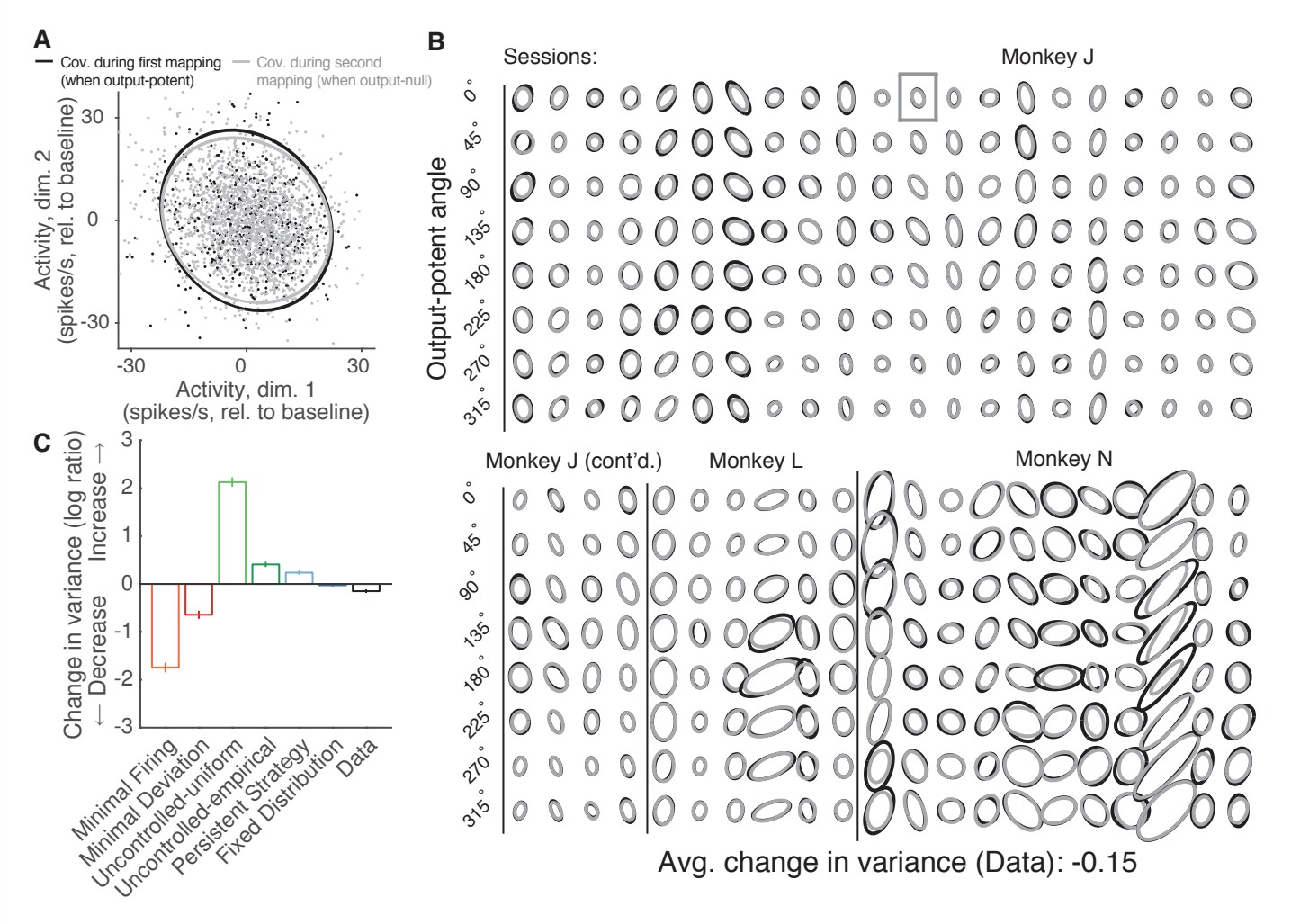

**Figure 6.** Variance of neural activity in dimensions that become output-null. (**A**) Observed activity from a representative session in the 2D subspace in which activity was output-potent under the first mapping and output-null under the second mapping. Activity recorded during use of the first mapping (black points) was output-potent while activity recorded during use of the second mapping (gray points) was output-null. The covariances during the first and second mapping (black and gray ellipses, respectively) are depicted as the 95% contours of a Gaussian density fit to the activity. Session J20120403, for all time steps when the activity would have moved the cursor to the right under the second mapping. (**B**) Covariance ellipses for all sessions and eight different cursor movement angles. Same conventions as in (**A**). Ellipses shown in (**A**) indicated by gray box. (**C**) Change in variance of neural activity in the same subspace as in (**A**), for the activity observed ('Data') and predicted by each hypothesis. Height of bars depicts the average change in variance across sessions (mean ± 2 SE).

DOI: https://doi.org/10.7554/eLife.36774.017

The following source data and figure supplements are available for figure 6:

**Source data 1.** Observed and predicted change in covariance across sessions, as depicted in *Figure 6C*.
DOI: https://doi.org/10.7554/eLife.36774.020

**Figure supplement 1.** Variance of neural activity did not change in the dimensions that became output-potent.
DOI: https://doi.org/10.7554/eLife.36774.018

**Figure supplement 2.** Decrease in variance was not accompanied by the mean activity moving toward predictions of minimal energy hypotheses.
DOI: https://doi.org/10.7554/eLife.36774.019

used previously to assess the errors of the hypotheses' predictions. Notably, binning in this way means that each bin may contain activity corresponding to different cursor movements under the *first* mapping, and so one might expect that in each bin the activity recorded under the first mapping would be more heterogeneous than the activity recorded under the second mapping.

We observed that the variance of population activity recorded under the first and second mappings was remarkably similar in the dimensions that changed from output-potent to output-null, even though these activity patterns usually corresponded to different cursor movements under the two mappings (*Figure 6B*). Thus, the variance of activity did not change much when an output-potent dimension became output-null, in agreement with the predictions of the Fixed Distribution hypothesis. To quantify these observations, we computed the average change in variance in each session (see Materials and methods). Across sessions, we found that the variance of observed activity showed a small but significant decrease when it became output-null (*Figure 6C*, 'Data') (t-test, $p<0.001$). This is in contrast to the predictions of the Minimal Firing and Minimal Deviation hypotheses, which predicted much larger decreases.

The observed change in variance lies closest to the predictions of the Fixed Distribution hypothesis. In fact, we observed that the Fixed Distribution hypothesis also predicted a slight decrease in variance in dimensions that became output-null (*Figure 6C*, 'Fixed Distribution') (t-test, $p<0.001$). This slight predicted change in variance occurs because the distributions of activity in the output-potent dimensions of the second mapping are different under the first and second mappings. Because the Fixed Distribution hypothesis predicts a fixed conditional distribution of output-null activity given the output-potent activity, slightly different sets of output-potent activity will result in a slightly different distribution of the corresponding output-null activity.

These analyses show that, contrary to the predictions of the minimal firing and uncontrolled hypotheses, the variance of population activity did not change dramatically in dimensions that were output-potent under the first mapping and output-null under the second mapping. We also assessed whether the reverse was true—if the variance of activity changed in dimensions that began as output-null and became output-potent. To measure this, we repeated the above analyses after predicting output-null activity produced during the first mapping using the activity observed under the second mapping (as in *Figure 5—figure supplement 2*). We found that the activity showed little to no change in variance in these dimensions (t-test, $p>0.5$), in agreement with the predictions of Fixed Distribution (*Figure 6—figure supplement 1*).

Importantly, the agreement between the observed output-null activity and the predictions of the Fixed Distribution hypothesis in these analyses indicates that our ability to accurately predict the distribution of output-null activity is not merely a result of most activity being output-null under both mappings. Instead, the distribution of output-null activity remains consistent with the Fixed Distribution hypothesis even in the output-null dimensions that were previously output-potent.

In *Figure 6C*, the observed output-null activity showed a larger decrease in variance than the predictions of the Fixed Distribution hypothesis, at least in the 2D subspace of output-null activity that was output-potent during the first mapping. This slight decrease in variance is in the direction of the predictions of Minimal Firing and Minimal Deviation. If this decrease in variance is to be explained by Minimal Firing or Minimal Deviation principles, we would expect that the observed *mean* output-null activity would also move in the direction of the predictions of Minimal Firing and Minimal Deviation, relative to what is predicted by Fixed Distribution. To see if this was the case, we first computed the distance of the observed mean output-null activity from the mean predicted by Minimal Deviation for each movement direction, and compared this to the distance of the mean output-null activity predicted by Fixed Distribution from the mean predictions of Minimal Deviation (*Figure 6—figure supplement 2A*). We did not find evidence that the observed mean output-null activity was closer to the mean predicted by Minimal Deviation than was the mean predicted by Fixed Distribution (one-sided Wilcoxon signed rank test, $p>0.5$; see *Figure 6—figure supplement 2B* and Materials and methods). Repeating the analysis with Minimal Firing instead of Minimal Deviation yielded similar results (one-sided Wilcoxon signed rank test, $p>0.5$). Thus, while we observed a slight decrease in the *variance* of output-null activity in dimensions that changed from output-potent to output-null, we did not find any evidence that the *mean* output-null activity moved in the direction of the predictions of Minimal Firing or Minimal Deviation.

## Discussion

Recent work has suggested that neural redundancy may be exploited for various computations (*Druckmann and Chklovskii, 2012*; *Kaufman et al., 2014*; *Moreno-Bote et al., 2014*; *Elsayed et al., 2016*; *Driscoll et al., 2017*; *Murray et al., 2017*). However, if the activity in output-null dimensions is constrained by the output-potent activity, then this may limit the ability of output-null activity to perform computations without affecting the readout. Here, we studied neural redundancy in the primary motor cortex using a BCI, where it is known exactly which population activity patterns are redundant, meaning they produce an identical cursor movement. We generated predictions of the distributions of output-null neural activity for subjects performing a BCI cursor control task, and compared them to the distributions observed in our experiments. We found that hypotheses inspired by minimal firing and minimal intervention principles, drawn from theories of muscle coordination, did not accurately predict the observed output-null activity. Instead, we found that the distribution of output-null activity was well predicted by the activity in the two output-potent dimensions. This coupling between the output-potent and output-null activity implies that, when output-potent activity is used to satisfy task demands, there are constraints on the extent to which neural circuits can use redundant activity to perform additional computations.

Our results indicate that the way in which neural redundancy is resolved is different from how muscle redundancy is resolved. There have been several prevalent proposals for how muscle redundancy is resolved, including minimal energy, optimal feedback control (OFC), and habitual control. Models incorporating minimal energy principles have helped to explain observed gait (*McNeill Alexander and McNeill, 2002*) and arm reaches (*Thoroughman and Shadmehr, 1999*; *Huang et al., 2012*; *Fagg et al., 2002*; *Farshchiansadegh et al., 2016*). By analogy, it has been proposed that the brain may prefer an 'economy of impulses' (*Barlow, 1969*; *Softky and Kammen, 1991*; *Levy and Baxter, 1996*), resolving neural redundancy by minimizing the production of action potentials. However, we found that minimal energy principles in terms of firing rates do not play a dominant role in the selection of output-null neural activity. Given that metabolic activity can decrease without corresponding changes in firing rates (*Picard et al., 2013*), the brain may implement minimal energy principles without influencing the way neural redundancy is resolved.

OFC posits that motor control signals are selected to minimize a cost function that depends on task requirements and other factors, such as effort or delayed reward. OFC models have been widely used to explain muscle activity during motor tasks (*Todorov, 2004*; *Scott, 2004*; *Diedrichsen et al., 2010*). Our results for neural activity differ in two important respects from OFC predictions with standard cost functions involving task requirements and effort. First, those implementations of OFC predict that variability in task-irrelevant dimensions should be higher than variability in task-relevant dimensions, a concept often referred to as the 'uncontrolled manifold' (*Scholz and Schöner, 1999*). We found that the variability of neural activity did not increase in dimensions that went from being task-relevant to task-irrelevant (*Figure 6C*). Second, those implementations of OFC predict a 'minimal intervention' strategy, whereby activity in task-relevant dimensions is corrected independently of activity in task-irrelevant dimensions (*Todorov and Jordan, 2002*; *Valero-Cuevas et al., 2009*; *Diedrichsen et al., 2010*). Three of the hypotheses we tested incorporate this minimal intervention principle: Uncontrolled-uniform, Uncontrolled-empirical, and Persistent Strategy. None of these hypotheses predicted neural activity in task-irrelevant dimensions as accurately as did the Fixed Distribution hypothesis, which predicts that the distributions of task-relevant and task-irrelevant activity are yoked. Overall, our work does not rule out the possibility that OFC is appropriate for predicting neural activity. First, it may be possible to design a cost function such that OFC predictions are consistent with the findings presented here. Second, one could consider applying OFC with the control signal being the input to M1 (e.g. PMd activity), rather than the control signal being M1 activity (as we have done here) or muscle activity (where OFC has been traditionally applied). This could induce coupling between the output-potent and output-null dimensions of the M1 activity, and thereby yield predictions that are consistent with the findings presented here.

It has also been proposed that muscle recruitment is habitual rather than optimal, such that muscle recruitment under altered dynamics is a rescaled version of that under normal control (*de Rugy et al., 2012*). The results for habitual control are similar to what we found for neural activity, in that (1) we could predict activity from previously observed activity, and (2) we observed a tight coupling

of the distributions of task-relevant and task-irrelevant activity (in contrast to minimal intervention). However, the results for habitual control are different from our findings in that we found that subjects appear to use the same distribution of activity in each of two different BCI mappings, whereas different (overlapping) subsets of muscle activation patterns were used under different conditions in *de Rugy et al. (2012)*.

Given how many dimensions of population activity there are (in this case, 10), it is somewhat surprising that conditioning on only the two output-potent dimensions could provide so much explanatory power for predicting the distribution in the remaining neural dimensions. This suggests that many of the dimensions of population activity are coupled, that is, changing the activity along some dimensions may also lead to changes along other dimensions, even though those dimensions are mutually orthogonal. During arm movement control, output dimensionality and presumably the neural dimensionality are larger than in our BCI setup. We speculate that during arm movements, many of the null dimensions will remain coupled with the potent dimensions, thereby yielding results similar to what we found here. Future work could examine whether animals can be trained to uncouple dimensions, as well as the effects of larger output-potent dimensionality on redundancy, by repeating our analyses with a higher-dimensional effector, such as a multiple degree-of-freedom robotic limb (e.g. *Wodlinger et al., 2015*).

The results presented here are related to, and go beyond, those in *Golub et al. (2018)*. Although the two studies analyzed data from the same experiments, they ask distinct questions. *Golub et al. (2018)* focused on explaining the changes in population activity underlying behavioral learning. By contrast, in the present work we seek to determine the constraints on activity in the task-irrelevant (i.e. output-null) dimensions. In other words, while *Golub et al. (2018)* focused on explaining the changes leading to *behavioral* learning, we focus here on the principles *other than behavior* that constrain population activity. As a result, all hypotheses we consider in the present work make predictions consistent with the observed behavior in the output-potent dimensions.

*Golub et al. (2018)* found that the amount of learning animals showed was consistent with a fixed neural repertoire of population activity patterns being reassociated to control the second BCI mapping. The repertoire of population activity refers to the set of population activity patterns that were observed, whereas here we focused on the *distribution*, which describes how often the animals produced different activity patterns. In other words, the finding of a fixed repertoire is a statement about the support of the distribution of population activity, whereas here we found that the distribution of population activity can be predicted in output-null dimensions, given the output-potent activity. Because many different distributions of neural activity can be constructed from a fixed repertoire, the present results represent a stronger constraint on population activity than that shown in *Golub et al. (2018)*. Indeed, the majority of the hypotheses we tested were consistent with a fixed neural repertoire, and thus cannot be disambiguated based on our prior work. This is evidenced by the predicted distributions largely overlapping with the support of the actual data distributions (*Figures 2–4*). The two hypotheses that were not fully consistent with a fixed repertoire are the Minimal Firing and Uncontrolled-uniform hypotheses. However, in the context of predicting the distribution of activity in redundant dimensions, these hypotheses represent interesting cases worth considering (i.e. where population activity either obeys minimal firing constraints, or that the output-null activity is fully unstructured, respectively), and so we included these hypotheses to cover these possibilities.

It is interesting to consider the relationship between arm movements and BCI cursor movements (*Orsborn et al., 2014*; *Vyas et al., 2018*). If the dimensions responsible for moving the arm overlap with both the output-potent and output-null dimensions of the BCI, this might explain the coupling we observe between the output-potent and output-null dimensions. However, in these experiments, the animal's arm was not moving during BCI control (see Extended Data Figure 5 in *Sadtler et al., 2014*). Thus, the activity we study here resides within the arm's output-null dimensions. This implies that in our recordings the arm's output-potent dimensions do not overlap with either the output-potent or the output-null dimensions of the BCI, and so arm movements (or the lack thereof) are unlikely to explain the coupling we observed between the output-potent and output-null dimensions of the BCI. Overall, being unaware of extra output-potent dimensions would likely make the predictions of the Fixed Distribution hypothesis worse, not better. The reason for this is as follows. The Fixed Distribution hypothesis predicts that the distribution of activity in output-null dimensions depends upon the corresponding output-potent activity. Under this hypothesis, the more we know of the output-potent activity, the better we can predict the output-null distribution. If there is an

output-potent dimension that we have not accounted for in our analyses, accounting for this dimension would likely improve our predictions. The fact that we were able to accurately predict the output-null distributions (13% histogram error on average, with the lowest possible error being 7%) without knowing all the potent dimensions is then evidence that these extra potent dimensions, if they exist, would not provide substantial additional predictive power.

In this work, we define a set of population activity patterns as redundant if they all result in the same readout in downstream areas. This definition of redundancy comes from early work on motor control (*Bernstein, 1967*; *Sporns and Edelman, 1993*), where it was noted that different motor signals can result in the same movement kinematics. This is related to but distinct from the information-theoretic definition of redundancy (*Schneidman et al., 2003*; *Latham et al., 2005*; *Averbeck et al., 2006*). In the information-theoretic case, redundancy describes the extent to which correlations among neurons limit decoding accuracy for different stimuli. This is distinct from the type of redundancy studied here, defined as the existence of multiple population activity patterns corresponding to the same readout. For example, by the information-theoretic definition, a system may have no redundancy (e.g. the population activity allows one to perfectly decode the encoded variable), but there may still be multiple population activity patterns that refer to this same encoded variable.

We found that the distribution of output-null activity could be well predicted using activity recorded under a different BCI mapping. Two factors of our experimental design are particularly relevant when interpreting this result. First, we used a balanced center-out task design in which subjects made roughly equal numbers of movements in each direction. If we had, for example, required far more leftward than rightward movements, this would have altered the distribution of joint activity and skewed the estimates of output-null activity during the second mapping. Second, this study focused on short timescales, where we predicted output-null activity within one to two hours of subjects learning a new BCI mapping. On this timescale, the motor system must be able to rapidly learn a variety of different mappings between neural activity and behavior, and thus, a variety of different sets of redundant activity. An interesting avenue for further research would be to determine if the constraints we observe on neural redundancy remain over longer timescales. Given repeated practice with the same BCI mapping across days and weeks (*Ganguly and Carmena, 2009*), it is possible that there are different and perhaps fewer constraints on neural redundancy than what we found here.

We have tested six specific hypotheses about how neural redundancy is resolved. These hypotheses cover a spectrum of how strongly the activity in output-null dimensions is constrained, with the minimal firing hypotheses being the most constrained, the minimal intervention hypotheses being the least constrained, and the Fixed Distribution hypothesis lying in between. Although the hypotheses we tested are not exhaustive, the best hypothesis (Fixed Distribution) yielded predictions of the distributions of output-null activity whose marginal histograms differed from the data by only 13% on average (*Figure 4F*), where we estimated the lowest error possible to be 7% on average. Further improvements to the prediction accuracy may be possible by incorporating additional constraints, such as dynamics (*Shenoy et al., 2013*). It should be stressed that our focus here was on predicting the *distribution* of output-null activity. Future work can assess whether output-null activity can be predicted on a time-step-by-time-step basis.

The central premise of the null space concept is that some aspects of neural activity are read out by downstream areas (output-potent) while other aspects are not (output-null) (*Kaufman et al., 2014*). This idea is related to the study of noise correlations, where it was recognized that activity fluctuations that lie outside of a stimulus encoding space (i.e. 'stimulus-null') are not detrimental to the stimulus information encoded by the neurons (*Averbeck et al., 2006*; *Moreno-Bote et al., 2014*). Studies have also shown that structuring neural activity in an appropriate null space can allow for multiplexing of different types of information (*Mante et al., 2013*; *Raposo et al., 2014*), as well as stable behavior (*Leonardo, 2005*; *Rokni et al., 2007*; *Ajemian et al., 2013*) and stable working memory (*Druckmann and Chklovskii, 2012*; *Murray et al., 2017*) in the presence of time-varying neural activity. Additionally, the existence of output-null dimensions in the motor system may facilitate motor learning (*Moorman et al., 2017*; *Ranganathan et al., 2013*; *Singh et al., 2016*) or allow for motor preparation (*Kaufman et al., 2014*; *Elsayed et al., 2016*) or novel feedback processing (*Stavisky et al., 2017*) without causing overt movement. Our work suggests that there may be limits on the extent to which output-null activity might be leveraged for neural computation. The coupling we observe between the distributions of output-null and output-potent activity suggests that

output-null activity is not modified independently of output-potent activity. This coupling may cause activity fluctuations in a stimulus-null space to influence the downstream readout, or limit one's ability to plan the next movement without influencing the current movement. Moving forward, an important direction for understanding the computations performed by different brain areas is to find out which aspects of the neural activity are read out (*Pagan et al., 2013*; *Kaufman et al., 2014*) and to understand how the dependencies like those identified in this study impact the computations being performed.

## Materials and methods

### Defining the mapping between neural activity and cursor movement

Experimental methods are described in detail in both *Sadtler et al. (2014)* and *Golub et al. (2018)*. Briefly, we recorded from the proximal arm region of primary motor cortex (M1) in three male Rhesus macaques using implanted 96-channel microelectrode arrays (Blackrock Microsystems). All animal care and handling procedures conformed to the NIH Guidelines for the Care And Use of Laboratory Animals and were approved by the University of Pittsburgh's Institutional Animal Care and Use Committee. The population spiking activity in each non-overlapping 45 ms bin was computed as the number of threshold crossings on each channel. In each session, 85–94 neural units were recorded (25 sessions from monkey J, six sessions from monkey L, 11 sessions from monkey N). These sessions were analyzed previously in *Golub et al. (2018)*. Data from monkeys J and L were first presented in *Sadtler et al. (2014)*. The average firing rate of the neural units per session was 50 ± 8, 42 ± 4, and 55 ± 14 spikes/s (mean ± s.d.) for monkeys J, L, and N, respectively.

Each session began with a block of calibration trials. The calibration procedure for monkey J involved either passive observation of cursor movement, or closed-loop BCI cursor control using the previous day's BCI mapping. For monkeys L and N, we used a closed-loop calibration procedure that gradually stepped from passive observation to closed-loop control, as described in *Sadtler et al. (2014)*. We then applied factor analysis (FA) to the spike counts recorded during these calibration trials to identify the 10D linear subspace (i.e. the 'intrinsic manifold') that captured dominant patterns of co-modulation across neural units (*Churchland et al., 2010*; *Harvey et al., 2012*; *Sadtler et al., 2014*; *Athalye et al., 2017*). We then estimated the factor activity, $\mathbf{z}_t \in \mathbb{R}^{10 \times 1}$, as the posterior expectation given the observed spike counts, $\mathbf{u}_t \in \mathbb{R}^{q \times 1}$, where $q$ is the number of neural units:

$$\mathbf{z}_t = L^\top \left( LL^\top + \Psi \right)^{-1} (\mathbf{u}_t - \mathbf{d}) \qquad (1)$$

Here, $L, \Psi$, and $\mathbf{d}$ are FA parameters estimated using the expectation-maximization algorithm, where $\Psi$ is constrained to be a diagonal matrix. The factor activity, $\mathbf{z}_t$, can be interpreted as a weighted combination of the activity of different neural units. We refer to $\mathbf{z}_t$ as a 'population activity pattern.'

We next orthonormalized $\mathbf{z}_t$ so that it had units of spike counts per time bin (*Yu et al., 2009*), using the following approach. In our FA model, $L$ defines a mapping from low-dimensional factor space to the higher-dimensional neural space. Because the columns of $L$ are not orthonormal, the factor activity does not have the same units (spikes counts per time bin) as the neural activity. However, we can fix this by finding an orthonormal basis for the columns of $L$ (*Yu et al., 2009*). To do this, we apply the singular value decomposition, yielding $L = USV^\top$, where $U \in \mathbb{R}^{q \times 10}$ and $V \in \mathbb{R}^{10 \times 10}$ have orthonormal columns and $S \in \mathbb{R}^{10 \times 10}$ is diagonal. Then, we can write $L\mathbf{z}_t = U(SV^\top \mathbf{z}_t) = U\tilde{\mathbf{z}}_t$. Because $U$ has orthonormal columns, $\tilde{\mathbf{z}}_t = SV^\top \mathbf{z}_t$ has the same units (spike counts per time bin) as $\mathbf{u}_t$. For notational simplicity, we refer to $\tilde{\mathbf{z}}_t$ as $\mathbf{z}_t$ throughout. The values in $\mathbf{z}_t$ appear larger than those expected for a single neuron because this value tends to grow with the total number of neural units.

Over the course of each experiment, animals used two different BCI mappings (see 'Behavioral task' below). Each BCI mapping translated the resulting moment-by-moment factor activity ($\mathbf{z}_t$) into a 2D cursor velocity ($\mathbf{v}_t$) using a Kalman filter:

$$\mathbf{v}_t = A\mathbf{v}_{t-1} + B\mathbf{z}_t + \mathbf{c} \qquad (2)$$

For the first BCI mapping, $A \in \mathbb{R}^{2 \times 2}$, $B \in \mathbb{R}^{2 \times 10}$, and $\mathbf{c} \in \mathbb{R}^{2 \times 1}$ were computed from the Kalman filter

parameters, estimated using the calibration trials. For the second BCI mapping, we changed the relationship between population activity and cursor movement by randomly permuting the elements of $z_t$ before applying *Equation 2*. This permutation procedure can be formulated so that *Equation 2* still applies to the second BCI mapping, but with an updated definition of $B$ (*Sadtler et al., 2014*).

## Behavioral task

Each animal performed an 8-target center-out task by modulating its M1 activity to control the velocity of a computer cursor. Each session involved two different BCI mappings. The first mapping was chosen to be intuitive for the animal to use. The animal used this first mapping for 200–400 trials, after which the mapping was changed abruptly to a second BCI mapping. The second mapping was initially difficult for the animal to use, and the animal was given 400–600 trials to learn to use the second mapping. Both mappings were chosen to be within the animal's instrinic manifold, mappings that we found in previous work could be readily learned within one session (*Sadtler et al., 2014*).

At the beginning of each trial, a cursor appeared in the center of the workspace, followed by the appearance of one of eight possible peripheral targets (chosen pseudorandomly). For the first 300 ms of the trial, the velocity of the cursor was fixed at zero. After this, the velocity of the cursor was controlled by the animal through the BCI mapping. If the animal acquired the peripheral target with the cursor within 7.5 s, he received a water reward, and the next trial began 200 ms after target acquisition. Otherwise, the trial ended, and the animal was given a 1.5 s time-out before the start of the next trial.

## Session and trial selection

The data analyzed in this study were part of a larger study involving learning two different types of BCI mapping changes: within-manifold perturbations (WMP) and outside-manifold perturbations (OMP) (*Sadtler et al., 2014*). We found that animals learned WMPs better than OMPs. Because we need animals to show stable cursor control under both mappings, we only analyzed WMP sessions in this study. Among the WMP sessions, we further selected those in which the animal learned stable control of the second mapping (42 selected and 12 discarded). This was important because performance with the second mapping was generally not as good as with the first mapping (*Figure 1—figure supplement 1*), and we wanted to ensure that any potential results were not due to incomplete learning of the second mapping (see also 'Internal model estimation' below). We further subselected from each session only those trials which exhibited stable behavioral performance, using a metric defined below. This was done to ensure that we were analyzing trials for which animals used a consistent strategy for selecting activity patterns.

We included sessions in which there existed a block of at least 100 consecutive trials that showed both substantial learning of the second mapping and consistent behavior. To identify trials showing substantial learning, we computed the running mean of the target acquisition time (on correct trials only), smoothed with a 100-trial boxcar shifted one trial at a time. The smoothed acquisition time for a trial corresponded to the average acquisition time within a 100-trial window centered on that trial. We then normalized these values so that 1 corresponded to the largest acquisition time in the first 50 trials using the second mapping, and 0 corresponded to the smallest acquisition time in the subsequent trials using the second mapping. We defined trials showing substantial learning as those with normalized acquisition times below 0.5. Next, to identify trials with consistent behavior, we computed the running variance of the target acquisition time. This was computed by taking the variance of the smoothed acquisition time above in a 100-trial boxcar, shifted one trial at a time. We then normalized these variances so that 1 corresponded to the largest variance in the first half of trials using the second mapping, and 0 corresponded to the smallest variance in any trial using the second mapping. We defined trials showing stable behavior as those with normalized variance below 0.5. We then identified blocks of consecutive trials that passed both of these criteria, joining blocks if they were separated by no more than 10 trials. We then selected the longest such block of at least 100 trials for our analyses. If no such block of trials was found, we excluded that session from our analyses. This procedure resulted in the 42 sessions across three monkeys that we included in our analyses.

We analyzed only successful trials. To avoid analyzing time steps with potentially idiosyncratic cursor control, we also ignored portions of the trial when the cursor was closer than 50 mm or more

than 125 mm away from the origin. We repeated our analyses without the latter exclusion and obtained quantitatively similar results.

## Internal model estimation

When an animal uses a BCI mapping, its internal conception of the BCI mapping can differ from the actual BCI mapping, even during proficient control (*Golub et al., 2015*). As a result, the animal's conception of output-potent versus output-null dimensions can be different from those defined by the actual BCI mapping. To control for this possibility, we evaluated our predictions based on the animal's internal conception of the output-null dimensions, rather than the actual output-null dimensions of the BCI mapping. This is particularly important for the second mapping, but we also did this for the first mapping. We used a method (Internal Model Estimation, IME) that we developed previously for estimating the animal's internal model of the BCI mapping (*Golub et al., 2015*), with the exception that here we apply the model directly to the factor activity ($\mathbf{z}_t$) as opposed to the neural activity ($\mathbf{u}_t$), as was done in *Golub et al. (2015)*.

The main idea of the IME framework is that the animal generates neural activity consistent with aiming straight to the target through an internal model of the BCI mapping. Due to natural visual feedback delay, the animal cannot exactly know the current cursor position, and thus aims from an internal estimate of the current cursor position. The internal estimate of the cursor position is a feed-forward prediction based on previously issued neural activity and the most recently available visual feedback. *Figure 5—figure supplement 4A* shows a single-trial BCI cursor trajectory (black), along with the animal's internal belief (red 'whisker') about how cursor position (red dots) evolved from the cursor position known from the most recently available visual feedback. The final segments of the trajectories reflect the same neural activity, which produces the actual cursor velocity (black arrow) through the actual BCI mapping, or the animal's intended cursor velocity (red arrow) through the animal's internal model. The animal's velocity command viewed through the internal model points closer toward the target than the actual movement of the BCI cursor, corresponding to a smaller angular error. Across sessions, the animals' angular errors when using the second BCI mapping did not usually return to the original level of error that the animal achieved under the first mapping (*Sadtler et al., 2014*) (*Figure 5—figure supplement 4B*). However, when viewed through the animals' internal models of the BCI mappings, angular errors during the second mapping were more similar to those observed during the first mapping (*Figure 5—figure supplement 4C*). Thus, the internal model helps to control for possible incomplete learning of the second mapping.

We used IME to obtain the animal's internal model of the BCI mapping (in the form of $A, B, \mathbf{c}$ in *Equation 2*), which yielded a corresponding set of cursor velocities ($\mathbf{v}_t$), cursor-target angles ($\theta_t$), and bases for the output-potent and output-null dimensions of each mapping (see $N$ and $R$ below) that we used in our offline analyses. The results reported in the main text are based on these quantities obtained from IME. When we analyzed the data without using IME (i.e. using the actual output-null dimensions of the BCI mapping), all of the results we report still held (*Figure 5—figure supplement 3*).

## Defining output-null activity

In *Equation 2*, the matrix $B \in \mathbb{R}^{2 \times 10}$ linearly projects a 10-dimensional input (factor activity) to a 2-dimensional output (cursor velocity). Thus, for any given cursor velocity ($\mathbf{v}_t$) there are multiple values of factor activity ($\mathbf{z}_t$) that would produce it. These multiple values of factor activity are all behaviorally equivalent, and we refer to their existence as 'neural redundancy.'

Mathematically, it is useful to consider the null space, $Nul(B)$, and the row space, $Row(B)$, of the matrix $B$. The critical property of $Nul(B)$ is that for any element $\mathbf{y} \in Nul(B) \subseteq \mathbb{R}^{10}$, we have $B\mathbf{x} = B(\mathbf{x} + \mathbf{y})$ for all $\mathbf{x} \in \mathbb{R}^{10}$. In other words, any change in activity within the null space of $B$ has no effect on the cursor movement produced. On the other hand, to achieve a particular cursor velocity ($\mathbf{v}_t$), there is exactly one $\mathbf{x} \in Row(B)$ such that $B\mathbf{x} = \mathbf{v}_t$. Thus, the activity in the row space of $B$ uniquely determines the cursor movement. To find a basis for $Row(B)$ and $Nul(B)$, we took a singular value decomposition of $B = USV^T$, where the diagonal elements of $S$ were ordered so that only the first two values were nonzero. Then, we let $R \in \mathbb{R}^{10 \times 2}$ be the first two columns of $V$, and $N \in \mathbb{R}^{10 \times 8}$ be the remaining eight columns. The columns of $N$ and $R$ are mutually orthonormal and together form an orthonormal basis for the 10-dimensional space of factor activity. This allowed us to decompose the

factor activity $\mathbf{z}_t$ at each time step into two orthogonal components: (1) activity in the row space of $B$ that affects the cursor velocity, which we call the *output-potent* activity ($\mathbf{z}_t^r \in \mathbb{R}^2$); and (2) activity in the null space of $B$ that does not affect the cursor movement, which we call the *output-null* activity ($\mathbf{z}_t^n \in \mathbb{R}^8$):

$$\mathbf{z}_t = N\mathbf{z}_t^n + R\mathbf{z}_t^r$$
$$\text{where } \mathbf{z}_t^n := N^\top \mathbf{z}_t, \mathbf{z}_t^r := R^\top \mathbf{z}_t \tag{3}$$

Note that all behaviorally equivalent activity will have the same output-potent activity ($\mathbf{z}_t^r$), but can differ in output-null dimensions. Thus, for time steps with similar cursor movements, the subject's choice of 8D output-null activity ($\mathbf{z}_t^n$) describes how the subject selected activity from a set of behaviorally equivalent options. Because the cursor velocity ($\mathbf{v}_t$) at each time step is a combination of output-potent activity and the cursor velocity at the previous time step (see *Equation 2*), output-potent activity can be thought of as driving a change in the cursor velocity. Note that in the depictions of hypotheses in *Figure 1*, *Figure 2*, *Figure 3*, and *Figure 4*, we used $\mathbf{v}_t = B\mathbf{z}_t$ instead of *Equation 2* for clarity.

## Predicting output-null activity

Our goal for each experiment was to predict the distribution of observed output-null activity during the second mapping across time steps corresponding to a given cursor movement direction (defined as the angle of $\mathbf{v}_t$ in *Equation 2*). In the context of the center-out task, we assumed that cursor movements in the same direction but with different speeds were still behaviorally equivalent to the animal. This is supported by previous work that found substantially more direction-related information than speed-related information in both single-unit and population activity in M1 (*Golub et al., 2014*). For this reason we assessed the output-null distribution in bins of cursor movement direction rather than cursor velocity (i.e. direction × speed).

All hypotheses generated predictions of the distribution of output-null activity observed while animals used the second BCI mapping, unless otherwise noted. To generate predictions of the distributions of output-null activity, we made predictions of the output-null activity at each time step. This allowed us to ensure that our predictions were consistent with the cursor kinematics observed during the experiment. We then aggregated the predictions across all time steps during the experiment with a similar cursor movement direction. In all cases, the predicted output-null activity respected the intrinsic manifold (*Sadtler et al., 2014*), because the output-null activity lies in an 8-dimensional subspace of the 10-dimensional intrinsic manifold.

To generate a prediction of the output-null activity for a particular time step ($\mathbf{z}_t^n$), each hypothesis had access to three sources of information recorded during the experiments. First, all hypotheses used the observed output-potent activity ($\mathbf{z}_t^r$), in order to ensure that every prediction was physiologically plausible (see below). Second, all hypotheses except for the Minimal Firing hypothesis utilized factor activity recorded during use of the first BCI mapping to form their predictions of output-null activity. Finally, the Persistent Strategy hypothesis also utilized the current position of the cursor relative to the target, defined as the cursor-target angle ($\theta_t$).

We ensured that all predictions of output-null activity ($\hat{\mathbf{z}}_t^n$) corresponded to physiologically plausible neural activity ($\hat{\mathbf{u}}_t$). By 'physiologically plausible' we mean that the neural activity was non-negative, and no greater than the maximum number of spikes (per 45 ms time step) observed for that neural unit during trials using the first BCI mapping ($\mathbf{u}_{max}$). To enforce the constraint, we either incorporated the constraint $0 \le \hat{\mathbf{u}}_t \le \mathbf{u}_{max}$ directly in the optimization problem (Minimal Firing hypothesis), or rejected predictions of neural activity that fell outside of the constraint (all other hypotheses). In the latter case, we combined the predicted output-null activity with the observed output-potent activity at that time step to form the predicted factor activity ($\hat{\mathbf{z}}_t$). We then converted this value to neural activity using the FA generative model:

$$\hat{\mathbf{u}}_t := L\hat{\mathbf{z}}_t + \mathbf{d} \tag{4}$$

If this neural activity was not physiologically plausible, we attempted to generate a new prediction of $\hat{\mathbf{z}}_t^n$ according to the hypothesis. This was possible because all hypotheses incorporated some form of sampling to generate their predictions. If this procedure failed even after 100 attempts to

generate a physiologically plausible prediction, we skipped making a prediction for that time step. This happened for less than 1% of all time steps.

## Minimal firing hypotheses

According to the Minimal Firing hypothesis, generating spikes incurs a metabolic cost. Thus, the subject should select the population activity pattern that involves the fewest spikes among all patterns that generate the desired cursor movement. Predictions for this hypothesis were generated as follows. For each time step, we find the spiking activity closest to zero firing that produces the observed cursor velocity:

$$\hat{\mathbf{u}}_t := \arg\min_{u} \| \mathbf{u} \|_2^2$$
$$\text{subject to } \mathbf{v}_t = A\mathbf{v}_{t-1} + Bf(\mathbf{u}) + \mathbf{c} \tag{5}$$
$$\text{and } 0 \leq \mathbf{u} \leq \mathbf{u}_{max}$$

Above, $f(\mathbf{u})$ refers to the factor activity corresponding to $\mathbf{u}$, as in **Equation 1**. Because $f(\mathbf{u})$ is a linear function of $\mathbf{u}$, the above minimization is a convex problem. $\mathbf{u}_{max}$ is the maximum activity level observed for each neuron, as described above. We solved for $\hat{\mathbf{u}}_t$ at each time step $t$ using Matlab's quadprog. All trends in results were the same if the $\ell_2$ norm in the optimization problem was changed to an $\ell_1$ norm. After solving the above minimization, we incorporated variability in spike generation by sampling from a Poisson: $\hat{\mathbf{u}}_t' \sim \text{Poisson}(\hat{\mathbf{u}}_t)$. We repeated this last step if necessary until $\hat{\mathbf{u}}_t'$ was physiologically plausible. Finally, we converted the prediction to factor activity, so that the resulting prediction of $\hat{\mathbf{z}}_t^n$ was $\hat{\mathbf{z}}_t^n := N^\top f(\hat{\mathbf{u}}_t')$.

We chose to incorporate Poisson variability into the predictions of the Minimal Firing (above) and Minimal Deviation hypotheses (below), rather than the Gaussian noise assumed by our FA model. The observed spike counts are discrete, whereas adding Gaussian noise would make the spike counts predicted by these hypotheses continuous. For this reason, to ensure a fair comparison we used Poisson variability, which will ensure the predictions remain discrete even after adding variability.

For the Minimal Deviation hypothesis, we generalized the Minimal Firing hypothesis so that instead of predicting the spiking activity nearest zero spikes/s, we predicted factor activity closest to some unknown activity level $\eta \in \mathbb{R}^{10}$. Solving this problem in the 10-dimensional factor space for the optimal value of $\eta$ yields lower prediction error than doing so in the $q$-dimensional neural space because we ultimately evaluate the hypotheses' predictions in factor space. After choosing $\eta$ (see below), the predicted factor activity was obtained by solving the following optimization problem:

$$\hat{\mathbf{z}}_t := \arg\min_{z} \| \mathbf{z} - \eta \|_2^2$$
$$\text{subject to } \mathbf{v}_t = A\mathbf{v}_{t-1} + B\mathbf{z} + \mathbf{c} \tag{6}$$

The above problem is known as a 'minimum norm' problem, and it turns out that the resulting solution's output-null activity, $\hat{\mathbf{z}}_t^n$, is a constant, for all $t$:

$$\hat{\mathbf{z}}_t^n = N^\top \eta \tag{7}$$

Because of the simple form of this solution, it was possible to choose the best value of $\eta$ for each session by minimizing the resulting output-null prediction error across cursor directions (see 'Error in mean' below). This value is:

$$\eta := N\left( \frac{1}{8} \sum_{i=1}^{8} \mu_i^n \right) \tag{8}$$

where $\mu_i^n$ is the average output-null activity in the $i^{th}$ cursor direction bin, which we estimate using activity recorded during the first BCI mapping. This ensures that the data used to evaluate the predictions were not used to obtain $\eta$. Finally, we incorporated spiking variability just as we did for the Minimal Firing hypothesis. To do this, we first converted the above prediction ($\hat{\mathbf{z}}_t$) to neural activity using the FA generative model ($\hat{\mathbf{u}}_t := L\hat{\mathbf{z}}_t + \mathbf{d}$). We then incorporated Poisson variability as described above, repeating the procedure until the resulting prediction was physiologically plausible, where

the prediction of $\hat{\mathbf{z}}_t^n$ was $\hat{\mathbf{z}}_t^n := N^\top f(\hat{\mathbf{u}}_t')$, with $\hat{\mathbf{u}}_t' \sim \mathrm{Poisson}(\hat{\mathbf{u}}_t)$.

## Uncontrolled-uniform hypothesis

According to the uncontrolled manifold concept (*Scholz and Schöner, 1999*), variability in output-null dimensions will be higher than that in output-potent dimensions. One explanation of this idea is the minimal intervention principle (*Todorov and Jordan, 2002*; *Valero-Cuevas et al., 2009*; *Diedrichsen et al., 2010*), which states that the variability in output-potent dimensions is controlled independently of the output-null activity, with the output-null activity being unmodified. While this principle specifies that output-null activity is independent of output-potent activity, it does not specify what the distribution of output-null activity actually is. Thus, we considered two hypotheses about this distribution. First, we supposed that the output-null activity would be uniformly distributed within bounds determined by the physiological range of population activity. This hypothesis thus predicts that activity in output-null dimensions has maximal entropy within the physiological range. For each $t$, we sampled:

$$\hat{\mathbf{z}}_t^n \sim \mathrm{Uniform}\left(\mathbf{z}_{min}^n, \mathbf{z}_{max}^n\right) \tag{9}$$

Above, $\mathbf{z}_{min}^n$ and $\mathbf{z}_{max}^n$ set the range on the minimum and maximum possible output-null activity. These bounds were set using population activity recorded during use of the first BCI mapping. We then resampled if necessary until our predictions generated physiologically plausible spiking activity when combined with the output-potent activity.

Note that in *Figure 2*, *Figure 3*, and *Figure 4* we applied PCA to the observed output-null activity to depict the three output-null dimensions with the most shared variance in the observed activity. Because of this, our visualizations of the distributions predicted by the Uncontrolled-uniform hypothesis in *Figure 3* appear mound-shaped rather than uniform. To understand this, suppose we sample from a uniform distribution over a rectangle in 2D. If we rotate this rectangle slightly, and visualize the distribution of points along the x-axis, the distribution will be mound-shaped. Similarly, the Uncontrolled-uniform hypothesis samples from a uniform distribution in the 8-dimensional output-null space, where the bounds of the rectangle are determined by bf $\mathbf{z}_{min}^n$ and $\mathbf{z}_{max}^n$ above. Applying PCA rotates this activity, such that the density along the PC dimensions appears mound-shaped.

## Uncontrolled-empirical hypothesis

Next, we considered a different hypothesis about the distribution of output-null activity under the minimal intervention principle. Rather than assuming output-null activity is uniformly distributed, we obtained an empirical distribution using population activity observed under the first mapping. To produce predictions of output-null activity during the second mapping, for each time step during the second mapping we sampled randomly from the population activity observed under the first mapping, and assessed the projection of that activity in the null space of the second mapping.

Concretely, let $T_1$ be the set of all time steps under the first mapping, and $T_2$ be the set of all time steps under the second mapping. Our prediction for each $t \in T_2$ is obtained by randomly sampling with replacement:

$$\hat{\mathbf{z}}_t^{\mathbf{n}} \sim \left\{ N^\top \mathbf{z}_i \,|\, \forall i \in T_1 \right\} \tag{10}$$

In other words, at each time step using the second mapping, we randomly select factor activity observed during the first mapping ($\mathbf{z}_i$), and project it into the null space of the second mapping ($N^\top \mathbf{z}_i$). We then resampled if necessary until our predictions generated physiologically plausible spiking activity when combined with the output-potent activity.

## Persistent Strategy hypothesis

An extension of the Uncontrolled-empirical hypothesis is motivated by the idea that the subject may select activity under one mapping by modifying the activity he used under the first mapping. For a given cursor-target angle, if the subject selects the same population activity under the second mapping as under the first mapping, that activity may not move the cursor towards the target under the second mapping. To correct the cursor movement, he modifies this activity according to the minimal intervention principle (*Todorov and Jordan, 2002*; *Valero-Cuevas et al., 2009*; *Diedrichsen et al.,*

*2010*), correcting activity only along output-potent dimensions of the current mapping. Concretely, for each $t \in T_2$, we sampled with replacement:

$$\hat{\mathbf{z}}_t^{\mathbf{n}} \sim \left\{ N^\top \mathbf{z}_i | \forall i \in T_1 \text{ such that } \theta_i \in \theta_t \pm 22.5° \right\} \tag{11}$$

where $\theta_t$ is the cursor-to-target angle at time $t$. As before, we resampled if necessary until our predictions generated physiologically plausible spiking activity when combined with the output-potent activity. This hypothesis is identical to the Uncontrolled-empirical hypothesis, except that at each time step we sampled only from time steps during the first mapping that had a similar cursor-target angle (i.e. within a 45° wedge around $\theta_t$). We found no consistent improvements in varying the constraints on the cursor-target angle (i.e. using values other than 22.5° in *Equation 11*), or when using the output-potent angle rather than the cursor-target angle.

## Fixed Distribution hypothesis

According to the Fixed Distribution hypothesis, the activity in output-null dimensions is tightly coupled to the activity in output-potent dimensions, even under different BCI mappings when these dimensions are not necessarily still null and potent. This is in contrast to the three previous hypotheses (Uncontrolled-uniform, Uncontrolled-empirical, Persistent Strategy), which all incorporated a minimal intervention principle, whereby output-null activity can be modified independently of the output-potent activity, within the physiological limits on the firing rate of each unit.

Under the Fixed Distribution hypothesis, we predict that the distribution of output-null activity given the output-potent activity will be the same distribution as it was under the previous mapping. To implement this hypothesis, for each time step during the second mapping, we predict that the subject selects whichever activity pattern he produced under the previous mapping that would best match the current output-potent activity. Specifically, given the output-potent activity produced during the second mapping ($\mathbf{z}_t^r$), we found the time step during the first mapping ($i^* \in T_1$) where the factor activity $\mathbf{z}_{i^*}$ would have come closest to producing that output-potent activity using the second mapping. Our prediction for output-null activity was then the output-null component of $\mathbf{z}_{i^*}$ through the second mapping ($N^T \mathbf{z}_{i^*}$). Mathematically, for each $t \in T_2$ our prediction was:

$$\hat{\mathbf{z}}_t^{\mathbf{n}} := N^\top \mathbf{z}_{i^*}$$
$$\text{where } i^* = \arg\min_{i \in T_1} \| \mathbf{z}_t^r - R^\top \mathbf{z}_i \|_2^2 \tag{12}$$

We observed that these predictions all satisfied physiological constraints, which suggests that the values of $\mathbf{z}_{i^*}$ selected at each time step each produced output-potent activity sufficiently close to $\mathbf{z}_t^r$.

This above implementation is also equivalent to the following: At each time step we identified the $K$ previously observed population activity patterns that would produce output-potent activity closest to the current output-potent activity under the second mapping. We then selected one of these patterns at random, and used the output-null activity of that pattern as our prediction at that time step. In our above implementation, $K = 1$. We found that using other values of $K$ (e.g. $K = 50$, $K = 200$) yielded similar results.

## Evaluating predictions

For each session, we evaluated the predicted output-null distributions of the above hypotheses in terms of how well they matched the observed output-null distributions for all time steps with similar cursor movements. To do this, we first grouped time steps by their corresponding cursor velocity into eight non-overlapping bins of cursor movement directions $(0° \pm 22.5°, 45° \pm 22.5°, ..., 315° \pm 22.5°)$. We then evaluated the accuracy of the predictions for each cursor movement direction.

For consistency, all predictions were evaluated in terms of factor activity. The Minimal Firing hypothesis generated its predictions in terms of neural activity, and we converted these predictions to factor activity using *Equation 1*.

## Histogram overlap

We compared the predicted and observed distributions of output-null activity in each dimension in terms of the average overlap of their histograms. For each session, we selected a single bin size for

all histograms using cross-validation (*Rudemo, 1982*). Then, for each cursor direction and output-null dimension, we computed the error between the observed ($\mathbf{y}$) and predicted ($\hat{\mathbf{y}}$) histograms. Let $\mathbf{y}_i$ be the normalized frequency in the $i^{th}$ bin, so that $\sum_{i=1}^{m} \mathbf{y}_i = 1$, and similarly for $\hat{\mathbf{y}}$. Then the histogram error was computed as follows:

$$L(\mathbf{y}, \hat{\mathbf{y}}) = \frac{1}{2} \sum_{i=1}^{m} |\mathbf{y}_i - \hat{\mathbf{y}}_i| \tag{13}$$

Above, $\frac{1}{2}$ is included so that $L(\mathbf{y}, \hat{\mathbf{y}}) = 1$ if the two histograms are completely non-overlapping. $L(\mathbf{y}, \hat{\mathbf{y}}) = 0$ if the two histograms are identical. This error was then averaged across all cursor directions and output-null dimensions. We multiplied this value by 100 to yield the average histogram error percentages reported in the main text.

For the visualizations in *Figure 2*, *Figure 3*, and *Figure 4*, we displayed the marginal histograms in the three output-null dimensions with highest variance in the observed output-null activity, as found by PCA. For all error calculations we considered all eight output-null dimensions without applying PCA.

## Error in mean

We assessed how well our predictions matched the observed mean output-null activity for each cursor movement direction. For all time steps in the same cursor movement direction bin, let $\mu^{\mathbf{n}} \in \mathbb{R}^{8 \times 1}$ be the vector of the mean observed output-null activity, and $\hat{\mu}^{\mathbf{n}} \in \mathbb{R}^{8 \times 1}$ the mean output-null activity predicted by a particular hypothesis. These are both vectors, and so we computed the distance between them using the $\ell_2$ norm:

$$L(\mu^{\mathbf{n}}, \hat{\mu}^{\mathbf{n}}) = \| \mu^{\mathbf{n}} - \hat{\mu}^{\mathbf{n}} \|_2 \tag{14}$$

For each hypothesis, we computed the error in mean in each cursor movement direction bin, and took the average of these values as the error in mean for each session.

## Error in covariance

We next assessed how well our predictions matched the observed covariance of output-null activity for each cursor movement direction. Let $C^n \in \mathbb{R}^{8 \times 8}$ and $\hat{C}^n \in \mathbb{R}^{8 \times 8}$ be the covariance of the observed and predicted output-null activity, respectively. There are a variety of methods for comparing covariance matrices, such as comparing their trace or determinant. We chose a metric invariant to affine transformations (e.g. scaling, translations, rotations) of the coordinate system (*Dryden et al., 2009*). Because the amount of variance in the recorded population activity might vary from session to session, this property of affine invariance helps ensure we can reasonably compare our covariance errors across sessions.

Let $\lambda_i \left( C^n, \hat{C}^n \right)$ be the $i^{th}$ generalized eigenvalue of $C^n$ and $\hat{C}^n$ (i.e. a value $\lambda$ such that $\det \left( C^n - \lambda \hat{C}^n \right) = 0$). Then following *Lang (1999)* and *Förstner and Moonen (2003)*, we computed the distance between these two matrices as:

$$L\left( C^n, \hat{C}^n \right) = \sqrt{\sum_i \log^2 \lambda_i \left( C^n, \hat{C}^n \right)} \tag{15}$$

If $C^n = \hat{C}^n$, then $L\left( C^n, \hat{C}^n \right) = 0$. For each hypothesis, we computed the error in covariance in each cursor movement direction bin, and took the average of these values as the error in covariance for each session.

## Error floor

To estimate the smallest errors achievable by any hypothesis (the 'error floor'), given a limited number of samples to estimate the true output-null distributions, we performed the following analysis. For each session, we randomly split the data during the second mapping in half, and measured the histogram, mean, and covariance errors when using the output-null activity from one half to predict the distribution of the output-null activity during the other half. We repeated this process 100 times

per session, and took the averages of the resulting errors as our estimates of the error floors for that session.

## Activity that became output-null in the second mapping

We sought to assess whether the variance of population activity changed in dimensions that became output-null under the second mapping. To do this, we identified the subspace of activity that was output-potent under the first mapping, but output-null under the second mapping.

As before, let the columns of $N$ be a basis for the null space of the second mapping. Now let the columns of $R_1$ be a basis for the row space of the first mapping. Then the space spanned by the columns of $(NN^\top)R_1 \in \mathbb{R}^{10 \times 2}$ describes the activity that would move the cursor during the first mapping but would not move the cursor during the second mapping. Let $S \in \mathbb{R}^{10 \times 2}$ be an orthonormal basis for $(NN^\top)R_1$, which we obtained by performing a singular value decomposition. Now let $Z \in \mathbb{R}^{10 \times n}$ be a matrix of $n$ factor activity patterns. To measure the amount of variance of $Z$ in the subspace spanned by the columns of $S$, we computed $\mathrm{Trace}(\mathrm{Cov}(Z^\top S)) \in \mathbb{R}$.

To assess how the variance of activity changes when it becomes irrelevant to cursor control, we grouped the time steps based on the cursor movement angle under the *second* mapping, for activity recorded under both the first and second mappings. First conditioning on the movement angle under the second mapping is consistent with our earlier analyses, when comparing the predicted and observed output-null distributions. To compute the cursor movement angle through the second mapping for activity recorded under the first mapping, we used the terms of *Equation 2* not involving the cursor velocity at the previous time step (i.e. we computed $\mathbf{v}_t = B\mathbf{z}_t + \mathbf{c}$). For consistency, we recomputed the cursor movement angle for activity recorded under the second mapping in the same way.

Let $Z_1$ and $Z_2$ be the factor activity in the same cursor movement angle bin recorded during the first and second mappings, respectively. We then computed the ratio of variance $R$ as follows:

$$R = \log\left( \frac{\mathrm{Trace}\big(\mathrm{Cov}\big(Z_2^\top S\big)\big)}{\mathrm{Trace}\big(\mathrm{Cov}\big(Z_1^\top S\big)\big)} \right) \tag{16}$$

The sign of $R$ specifies whether the variance of activity increased ($R>0$) or decreased ($R<0$) when that activity became irrelevant to cursor control under the second mapping. We took the average of this ratio across all cursor movement direction bins to compute a ratio for each session.

To compute this ratio for the predictions of our hypotheses, as in *Figure 6C*, we substituted $Z_2$ with the predictions of our hypotheses, i.e. by combining their predicted output-null activity with the observed output-potent activity under the second mapping.

We also repeated the above analyses on our predictions of output-null activity produced during the first mapping using the activity observed under the second mapping, as shown in *Figure 5—figure supplement 2* and *Figure 6—figure supplement 1*. This was done by swapping the roles of the first and second mappings in the above analysis description.

## Distances of mean output-null activity from Minimal Firing and Minimal Deviation

For each cursor direction on each session, we computed the distance from the mean observed output-null activity to the mean predicted by the Minimal Deviation hypothesis, where the distance was computed as the $\ell_2$ norm between the two 8D mean vectors. We then compared this distance to the distance between the mean predicted by Fixed Distribution and the mean predicted by Minimal Deviation (*Figure 6—figure supplement 2*). If the latter distance was consistently smaller than the former, this would be evidence that the observed mean output-null activity had moved towards the predictions of Minimal Deviation, relative to what was predicted by Fixed Distribution. We did not find evidence that this was the case (one-sided Wilcoxon signed rank test, $p>0.5$), suggesting that the mean observed output-null activity was not closer to Minimal Deviation than expected under Fixed Distribution. We repeated the same analysis using the mean predicted by Minimal Firing instead of Minimal Deviation, and reached the same results (one-sided Wilcoxon signed rank test, $p>0.5$).

## Acknowledgments

The authors would like to thank Wilsaan Joiner and Doug Ruff for their feedback on the manuscript. This work was supported by NIH R01 HD071686 (APB, BMY, and SMC), NSF NCS BCS1533672 (SMC, BMY, and APB), NSF CAREER award IOS1553252 (SMC), NIH CRCNS R01 NS105318 (BMY and APB), Craig H Neilsen Foundation 280028 (BMY, SMC, and APB), Simons Foundation 364994 (BMY), and Pennsylvania Department of Health Research Formula Grant SAP 4100077048 under the Commonwealth Universal Research Enhancement program (SMC and BMY).

## Additional information

### Funding

| Funder | Grant reference number | Author |
| --- | --- | --- |
| National Science Foundation | NCS BCS1533672 | Aaron P Batista<br>Byron M Yu<br>Steven M Chase |
| National Institutes of Health | R01 HD071686 | Aaron P Batista<br>Byron M Yu<br>Steven M Chase |
| National Science Foundation | Career award IOS1553252 | Steven M Chase |
| National Institutes of Health | CRCNS R01 NS105318 | Aaron P Batista<br>Byron M Yu |
| Craig H. Neilsen Foundation | 280028 | Aaron P Batista<br>Byron M Yu<br>Steven M Chase |
| Simons Foundation | 364994 | Byron M Yu |
| Pennsylvania Department of Health | Research Formula Grant SAP 4100077048 | Byron M Yu<br>Steven M Chase |

The funders had no role in study design, data collection and interpretation, or the decision to submit the work for publication.

### Author contributions

Jay A Hennig, Software, Formal analysis, Writing—original draft, Writing—review and editing; Matthew D Golub, Peter J Lund, Kristin M Quick, Conceptualization, Writing—review and editing; Patrick T Sadtler, Conceptualization, Writing—review and editing, Performed the animal experiments; Emily R Oby, Writing—review and editing, Performed the animal surgeries, Performed the animal experiments; Stephen I Ryu, Elizabeth C Tyler-Kabara, Writing—review and editing, Performed the animal surgeries; Aaron P Batista, Byron M Yu, Steven M Chase, Conceptualization, Supervision, Writing—original draft, Writing—review and editing

### Author ORCIDs

Jay A Hennig http://orcid.org/0000-0001-7982-8553
Matthew D Golub http://orcid.org/0000-0003-4508-0537
Byron M Yu https://orcid.org/0000-0003-2252-6938
Steven M Chase http://orcid.org/0000-0003-4450-6313

### Ethics

Animal experimentation: All animal handling procedures were approved by the University of Pittsburgh Institutional Animal Care and Use Committee (protocol #15096685) in accordance with NIH guidelines. All surgery was performed under general anesthesia and strictly sterile conditions, and every effort was made to minimize suffering.

### Decision letter and Author response

Decision letter https://doi.org/10.7554/eLife.36774.026

Author response https://doi.org/10.7554/eLife.36774.027

## Additional files

### Supplementary files

• Supplementary file 1. Table of statistical tests.
DOI: https://doi.org/10.7554/eLife.36774.021

• Transparent reporting form
DOI: https://doi.org/10.7554/eLife.36774.022

### Data availability

Source data files have been provided for Figures 2-6. Code for analysis has been made available at https://github.com/mobeets/neural-redundancy-elife2018, with an MIT open source license (copy archived at https://github.com/elifesciences-publications/neural-redundancy-elife2018).

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

# Appendix 1

DOI: https://doi.org/10.7554/eLife.36774.023

To understand how our results might change if we recorded from more neural units, we assessed the dimensionality and shared variance of population activity with a varying number of units (*Williamson et al., 2016*) (*Appendix 1—figure 1*). For each session, we fit factor analysis (FA) models (as defined in *Equation 1*) to subsets of varying numbers of units and identified the number of factors needed to maximize the cross-validated data likelihood. This resulted in estimates of the model parameters $L$ and $\Psi$. As in *Williamson et al. (2016)*, dimensionality was defined as the number of eigenvectors of $LL^\top$ needed to explain 95% of the shared variance. Concretely, if the eigenvalues of $LL^\top$ are $\lambda_1, \lambda_2, ..., \lambda_D$, then $d_{shared}$ is the smallest $J$ such that $\left(\sum_{i=1}^{J} \lambda_i\right) / \left(\sum_{i=1}^{D} \lambda_i\right) \geq 0.95$. Note that the absolute dimensionality depends on the method (% shared variance threshold) and criterion (threshold = 95%) used for assessing dimensionality. This is the same method used in *Williamson et al. (2016)*, but differs slightly from the method used in *Sadtler et al. (2014)*. We found that the dimensionality of the population activity increased with the number of units (*Appendix 1—figure 1A*).

As in *Williamson et al. (2016)*, we computed the percentage of each neural unit's activity variance that was shared with other recorded units (% shared variance). We calculated the average percent shared variance across neurons as follows:

$$\text{Percent shared variance for neuron } k = 100 \frac{L_k L_k^\top}{L_k L_k^\top + \Psi_k} \tag{17}$$

where $L_k$ is the row of $L$ corresponding to unit $k$. We found that the % shared variance initially increased with the number of units, then reached an asymptote, such that the % shared variance was similar with 30 and 85 units (*Appendix 1—figure 1B*). The results in *Appendix 1—figure 1A–B* imply that the top $\sim 10$ dimensions explain nearly all of the shared variance, and that additional dimensions identified by recording from more units explain only a small amount of additional shared variance. Thus, recording from more units beyond the $\sim 85$ units that we recorded in these experiments is not likely to reveal additional dimensions with substantial shared variance. We next measured the principal angles between modes identified using 30 units with those identified using 85 units (*Appendix 1—figure 1C*) (*Bjorck and Golub, 1973*). Modes were defined as the eigenvectors of the shared covariance matrices corresponding to units from the 30-unit set (i.e. the eigenvectors of $LL^\top$ where $L$ includes only the rows corresponding to the same 30 units). To restrict the analysis to the number of modes used to estimate the intrinsic manifold, only the ten modes explaining the most shared variance were included in the principal angle calculations. The small principal angles between modes identified using 30 and 85 units indicate that the dominant modes remained largely unchanged when using more units, in agreement with *Williamson et al. (2016)*. These modes define the intrinsic manifold, the space within which we perform all of our analyses in the current work. Thus, recording from more units beyond the $\sim 85$ units that we recorded in these experiments is not likely to substantially change the results reported in this work.

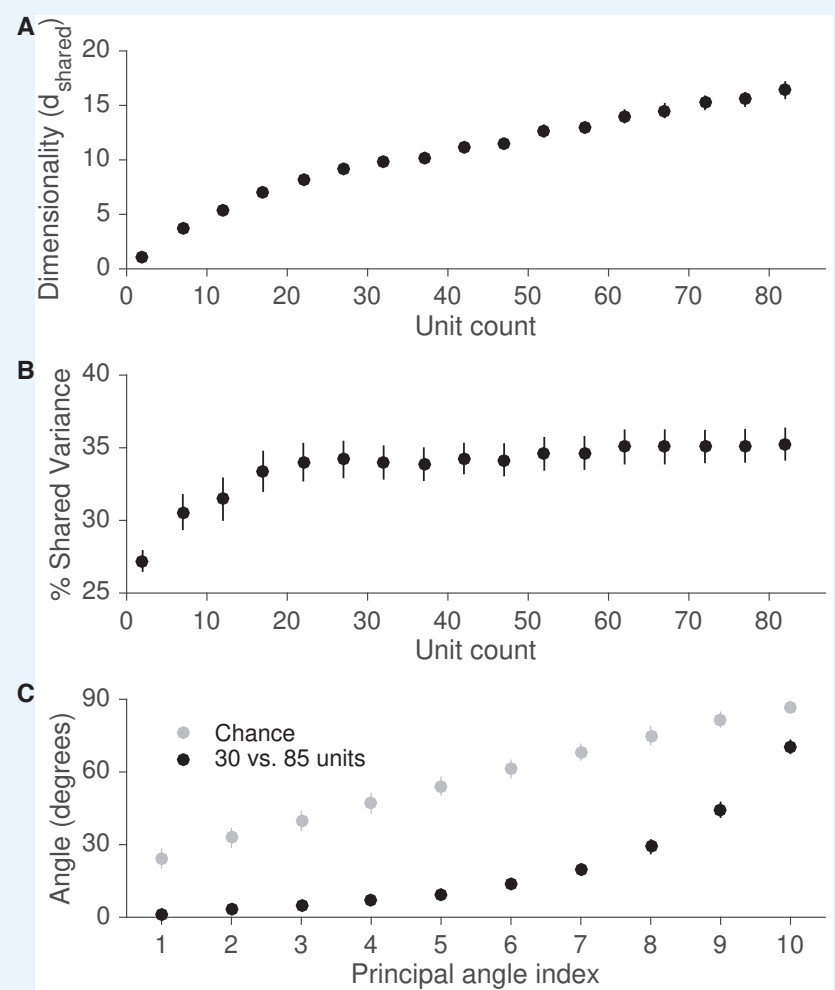

**Appendix 1—figure 1.** Recording from more units is likely to reveal an intrinsic manifold similar to that identified in this study. (**A**) We assessed the dimensionality ($d_{shared}$) of population activity after applying factor analysis to varying numbers of units from each session. Dimensionality is defined as the number of factors needed to explain 95% of the shared variance. Dimensionality increased with the number of units. Error bars depict mean $\pm$ SE, across sessions. (**B**) We also computed the percentage of each neural unit's activity variance that was shared with other recorded units (% shared variance). The % shared variance is based on the same factor analysis models identified in (**A**). The % shared variance initially increased with the number of units, then reached an asymptote, such that the % shared variance was similar with 30 and 85 units. Error bars depict mean $\pm$ SE, across sessions. (**C**) We next measured the principal angles between the modes identified by factor analysis using 30 units with those identified using 85 units. Modes are defined as the eigenvectors of the shared covariance matrices corresponding to units from the 30-unit set. The small principal angles between modes identified using 30 and 85 units indicate that the dominant modes remained largely unchanged when using more units. Gray points represent principal angles between random 30-dimensional vectors. Error bars for black points depict mean $\pm$ SE, across sessions, while error bars for gray points depict mean $\pm$ s.d.
DOI: https://doi.org/10.7554/eLife.36774.024

