## [Decision Letter]

Thank you for submitting your article "Constraints on neural redundancy" for consideration by *eLife*. Your article has been reviewed by three peer reviewers, including Eric Shea-Brown as the Reviewing Editor and Reviewer #1, and the evaluation has been overseen by Sabine Kastner as the Senior Editor. The following individual involved in review of your submission has agreed to reveal their identity: Guillaume Hennequin (Reviewer #2).

The reviewers have discussed the reviews with one another and the Reviewing Editor has drafted this decision to help you prepare a revised submission.

Summary:

In this paper, Hennig et al. investigate the structure of output-null activity as recorded during BCI tasks in which these output-null directions are under their experimental control. They go systematically through a series of hypotheses regarding the marginal distributions of population activity along the few relevant output-null directions (contained within a subspace that already accounts for most of the variance in the population activity). These hypotheses are both very clearly articulated (e.g. organised in 3 categories) and their quantitative assessment is thorough. The results argue against hypotheses that null-space neural variability is either uncontrolled or determined by energy constraints (the minimal firing and minimal deviation hypotheses, which are well explained and clearly linked). Rather the work shows that activity in potent and null dimensions is surprisingly coupled, which constrains activity in the redundant dimensions, at least over the hundreds of trials following a switch in the BCI decoder. They accomplish this by exploiting an experimental setup and dataset, previously studied in the context of other relearning questions, where monkeys are first trained to perform a cursor "reach" task and then are trained to perform the same task with a different mapping of activity to response.

The dataset and analyses that the paper uses are highly interesting, and the questions they address about adaptation and function of high-dimensional neural activity are very timely. Manifold perturbations and learning over them via BCI decoders is a setup where clear hypotheses on neural variability and covariability can directly be formulated and tested. The analysis has a high level of complexity but a set of statistical controls are carried out in the supplementary material, as well as in, e.g., Figure 6, to show that the analysis itself is robust. The metrics of histogram error and error floors are well motivated and explained. The writing is very clear and well explained. The hypotheses tested seemed appropriate and were well justified. Despite some concerns about over-interpretation and one about novelty below, these features lend confidence in the authors' results.

Essential revisions:

Several concerns arose about over-interpretation and novelty, as well as data that should be included. These need to be addressed, and are as follows:

A1) Golub et al. (2018), by many of the same authors and using a similar dataset (or perhaps exactly the same dataset; it isn't clear but should be) – is a closely related paper, in which the authors conclude that monkeys continue to produce a fixed set of neural activity patterns when BCI mappings change. In other words, Golub et al., 2018 argues that the same fixed set of activity patterns are reassociated with different BCI outputs. What is the impact of this published result, and the allied covariance analysis in Golub et al., on the a priori viability of the different hypotheses that the authors test in the present paper? For example, it seems consistent with the present fixed distribution hypothesis, but perhaps not with first version of the minimal firing hypothesis? The authors should do a much better job of discussing this and clarifying what the present paper adds – both throughout the paper and in the Discussion – rather than only giving a one sentence mention of Golub 2018 as in the current Discussion.

A2) The authors give no justification for this statement: "To fully utilize the proposed benefits of neural redundancy, there should be as few constraints as possible on which population activity patterns can be produced while leaving the readout unaffected."

Related to this: The authors state that "the output-null space should not be thought of as a space in which neural activity can freely vary to carry out computations that are not reflected in output-potent activity. Instead, the output-potent component of the population activity pattern determines the distribution of the corresponding output-null activity." Do the results support this conclusion? The results affirm that the covariance structure of neurons within the manifold is persistent across their experimental tasks. But does this mean that the null activity is not used for computation or preparation?

A3) The paper is based on a center-out BCI task. However, there were no figures or data included on the behavioral performance. After reading the methods, I saw a paper referenced. It would be nice to include a figure on behavioral performance in the current manuscript, especially given that a third monkey appears to have been added following the previously published paper. It is helpful to the reader to have the behavioral data present in the current paper rather than having to go back to the previous work.

The following points require clarification or rewriting in the main text and/or addition to Discussion:

B1) "Minimal intervention principle" (MIP, in this review) is used to describe a set of alternatives that – for us at least – do not really embody this notion. Here, the authors refer to MIP as the neural activity not moving along directions that do not matter for cursor movement, i.e. output-null directions. This view might stem from the authors considering the cursor to be the "plant", and M1 being the input to the plant. Another, perhaps more brain-centric view of the BCI control problem is to view both M1 and the cursor as the plant under control. After all, the brain needs to control the activity of M1 as part of controlling the cursor. Under this perspective, optimal control with MIP prescribes driving M1 with such minimal inputs as to adequately steer the activity in the directions that matter (output-potent) while being tolerant of fluctuations in output-null directions. Typically, though, such control inputs will generate activity trajectories that visit both subspaces in a coordinated way. For example, it might be that, due to the internal dynamics of M1, the most efficient way of moving its activity along an output-potent direction is to, in fact, make a pronounced excursion along an output-null direction. We are by no means rejecting hypotheses 3-5 as being irrelevant, but we think that MIP might in fact have more in common with hypothesis 6 ("fixed strategy") in that optimal control of a few activity directions is likely to cause correlated activity across potent- and null-directions due to the intrinsic dynamics of the circuit.

B2) Concerning the last hypothesis (fixed distribution), we believe it could be tested using any relevant distribution of activity, not only the one recorded under the "previous mapping"; it seems that all that matters is that you have a "reference distribution" that predicts some degree of coupling/correlations between the two sets of potent/null directions that define the current mapping. This, for example, could be the activity used to define the "intrinsic manifold". Have the authors considered testing this last hypothesis using such prior data? If it explains the structure of output-null activity equally well, this could add to the argument that output-null activity is mainly constrained by the dynamics of the circuit under control. The authors should comment on this, explain how this would fits into the results in the paper as a whole.

Related to this, it seems possible that the motor cortex activity could drive muscle movements, in addition to the BCI cursor. In this case, some of the output null dimensions could actually be related to output potent dimensions, just not for cursor movement. In this case, the so-called redundant or output null dimensions might not actually be redundant. Rather, they would be serving a different purpose, which could explain possible structure in their activity distributions. In this sense, I would not consider these redundant dimensions. Rather, the activity could be driving movements in a higher than 2D space if one considers both the BCI output and muscle movements. It would be helpful to see some description of how the authors have ruled out this possibility or why they have considered it to be a non-significant issue.

B3) Are the authors confident that, following a switch in the decoder, the animal has enough time with the new decoder for any expected null space changes to occur? The trial count into the hundreds does seem substantial, so this is not a big concern, though the authors might wish to comment on how long the animal had with the new decoder (especially whether this was over one or multiple sessions/days), in relationship to timescales of plasticity and expected changes in activity, in the Discussion.

B4) The BCI case is quite nice for studying this type of question. However, the dimensionality of the neural population controlling the BCI output is rather limited compared to perhaps a natural case of an entire motor cortical region capable of controlling a set of muscles. Dimensionality seems like it should play a significant role in considerations of redundancy, both on the neural end and the output end. It would be helpful to see more analysis or discussion of how neural and behavioral dimensionality, and perhaps most significantly the difference between the two, could play a role in interpretation and analysis of redundancy. For example, intuitively it seems possible that redundancy might be less constrained if 10,000 electrodes were used for BCI rather than 96. Is this true?

B5) The authors limit their analysis to the in-manifold perturbations of their experiment. The dataset has also out-of-manifold perturbations. Some comments on what is to be expected here seem worth mentioning in the Discussion.

---

## [Author Response]

Essential revisions:Several concerns arose about over-interpretation and novelty, as well as data that should be included. These need to be addressed, and are as follows:A1) Golub et al. (2018), by many of the same authors and using a similar dataset (or perhaps exactly the same dataset; it isn't clear but should be) – is a closely related paper, in which the authors conclude that monkeys continue to produce a fixed set of neural activity patterns when BCI mappings change. In other words, Golub et al., 2018 argues that the same fixed set of activity patterns are reassociated with different BCI outputs. What is the impact of this published result, and the allied covariance analysis in Golub et al., on the a priori viability of the different hypotheses that the authors test in the present paper? For example, it seems consistent with the present fixed distribution hypothesis, but perhaps not with first version of the minimal firing hypothesis? The authors should do a much better job of discussing this and clarifying what the present paper adds – both throughout the paper and in the Discussion – rather than only giving a one sentence mention of Golub 2018 as in the current Discussion.

We agree with the suggestion entirely. As the reviewers describe, Golub et al. (2018) found evidence that the amount of learning in these BCI experiments was consistent with a fixed “neural repertoire”, whereby neural activity patterns were reassociated (“Reassociation”) to control the second BCI mapping. In the next few paragraphs we discuss the ways in which the results in the present work go beyond the results of Golub 2018, as well as how the hypotheses we test are consistent with Golub et al.’s finding of a fixed neural repertoire.

The analyses in the present work were developed in parallel with those of Golub 2018, and while we analyzed data from the same experiments, the two papers ask distinct questions. In Golub 2018, the focus is on the amount of learning observed in these experiments, and how changes in the structure of population activity enabled that learning. By contrast, in the present work we seek to determine the constraints on activity in the task-irrelevant (i.e. output-null) dimensions. In other words, while Golub 2018 focused on explaining the changes leading to *behavioral* learning, we focus here on the principles *other than behavior* that constrain population activity. As a result, all hypotheses we consider in the present work make predictions consistent with the observed amount of learning in the output-potent dimensions.

Golub 2018 does not rule out or disambiguate among most of the hypotheses we test here. The *repertoire* of population activity describes the set of population activity patterns that were observed, while the *distribution* describes the frequencies with which these different patterns occurred. In other words, the repertoire describes the “support” of the distribution. The majority of the hypotheses we test are consistent with a fixed neural repertoire of population activity, as observed in Golub 2018. This is evidenced by the plots of output-null distributions during an example session (Figures 2-4), where the predicted distributions largely overlap with the support of the actual data distributions. The two hypotheses that, as it turns out, are not fully consistent with a fixed repertoire are the Minimal Firing and Uncontrolled-uniform hypotheses. However, in the context of predicting the distribution of activity in redundant dimensions, these hypotheses represent interesting cases that readers are likely to consider (i.e. where neural activity either obeys minimal firing constraints, or that the null activity is fully unstructured, respectively), and so we believe they are hypotheses worth including as useful points of reference.

The finding of a Fixed Distribution in the present work goes beyond the predictions of a fixed repertoire in the following ways. First, while the Reassociation hypothesis of Golub 2018 predicts a fixed repertoire of activity, it does not directly describe its distribution. This is an important distinction because many different distributions of neural activity can be constructed from a fixed repertoire. Here we find that the distribution of population activity in output-null dimensions can be predicted from the output-potent activity. This was a surprising discovery, even in light of Golub 2018. Second, Golub 2018 do find that the covariance of activity in the potent dimensions of the BCI mappings is conserved during control of the second mapping (Figure 5A in Golub 2018), similar to what we find in Figure 6C of the present manuscript. However, the present results go beyond this, as we find that not only is the *covariance* preserved in these 2D subspaces of neural activity, but in fact the *distribution* of activity in the 8D output-null dimensions is preserved. This is a stronger constraint than what Golub 2018 identified. This finding is important because there are many ways in which the covariance of activity in a 2D subspace could be preserved, including cases where the distribution in the 8D null space is different; knowing that in fact the distribution in the 8D null space is preserved suggests that output-null activity is much more constrained by the output-potent activity than one could have concluded from this previous work.

To summarize, the results of the present work are that the distribution of output-null activity is predicted by the activity in only the two potent dimensions. This indicates that the output-null activity is coupled with the output-potent activity, implying that the number of independently controllable degrees of freedom that a neural population can express may be even lower than the intrinsic dimensionality of the population activity. We also found that principles inspired by the study of muscular redundancy, namely minimal firing and minimal intervention, were not the best predictors for the selection of redundant neural activity. These results cannot be (or are not easily) deduced from the results of Golub 2018.

We have added the following paragraphs to the Discussion relating the present results with those reported in Golub 2018:

“The results presented here are related to, and go beyond, those in Golub et al. (2018). […] However, in the context of predicting the distribution of activity in redundant dimensions, these hypotheses represent interesting cases worth considering (i.e. where population activity either obeys minimal firing constraints, or that the output-null activity is fully unstructured, respectively), and so we included these hypotheses to cover these possibilities.”

We have also clarified in the Introduction of the manuscript that the results of Golub 2018 do not imply the results in the present work:

“We tested all hypotheses in terms of their ability to predict the distribution of output-null activity, given the output-potent activity. […] Therefore, to understand the principles governing the selection among redundant population activity patterns, we focused on predicting the *distribution* of redundant population activity within the intrinsic manifold and neural repertoire.”

We have also updated the Materials and methods to clarify that we analyzed data from the same experiments as in Golub et al. (2018).

“Experimental methods are described in detail in both Sadtler et al. (2014) and Golub et al. (2018)…In each session, 85-94 neural units were recorded (25 sessions from monkey J, 6 sessions from monkey L, 11 sessions from monkey N). These sessions were analyzed previously in Golub et al. (2018). Data from monkeys J and L were first presented in Sadtler et al. (2014).”

A2) The authors give no justification for this statement: "To fully utilize the proposed benefits of neural redundancy, there should be as few constraints as possible on which population activity patterns can be produced while leaving the readout unaffected."Related to this: The authors state that "the output-null space should not be thought of as a space in which neural activity can freely vary to carry out computations that are not reflected in output-potent activity. Instead, the output-potent component of the population activity pattern determines the distribution of the corresponding output-null activity." Do the results support this conclusion? The results affirm that the covariance structure of neurons within the manifold is persistent across their experimental tasks. But does this mean that the null activity is not used for computation or preparation?

These are all great points. To clarify what we meant by the statement in the Introduction, one proposed advantage of having redundancy in population activity is that it would allow the brain to perform computations without affecting readout. If there are constraints on this redundant activity (e.g., if task requirements on activity in the readout dimensions limit the redundant activity), this may limit the extent to which this activity is able to perform additional computations. We have rewritten the sentence in question so that this point is more clearly stated:

“To fully utilize the proposed benefits of neural redundancy, the population activity should be allowed to freely vary, as long as the readout of this activity remains consistent with task demands. This would allow the population activity to perform computations that are not reflected in the readout.”

We have clarified in the revised manuscript that our results do not imply that output-null activity is not used for computation (see below). Rather, we find that the output-potent activity likely constrains the extent to which output-null activity can be used for computation. Our work does not contradict previous findings (Kaufman et al., 2014; Elsayed et al., 2016) that computations can be carried out in the null space. Rather, we wish to prevent an overinterpretation of Kaufman and Elsayed, whereby readers might imagine that null space activity is completely available for computation. Our work suggests that the output-null distributions observed in these previous studies were limited or constrained by the different requirements for potent space activity during the two epochs in these studies (e.g., zero or constant potent activity during preparation, and non-zero or time-varying potent activity during execution).

In the other quoted sentence mentioned by the reviewers, we realized that the word “determines” was too strong and inadvertently implied that null space activity is not used for computation/preparation. Rather, we found that the distribution of null space activity was predicted by knowing the activity in potent/readout dimensions, indicating that output-potent and output-null activity do not vary independently. If the output-potent activity needs to be a certain value due to task demands, this can constrain how the output-null activity can vary, which likely constrains the computations that can be carried out in the output-null space. We have changed the last two sentences of the Introduction to clarify that our results suggest neural activity does not carry out null space computations *independently* of the output-potent activity, and we no longer use the word “determines”:

“Furthermore, the output-null space should not be thought of as a space in which neural activity can freely vary to carry out computations without regard to the output-potent activity. […] If the required output-potent activity is defined by the task demands, this can constrain how the output-null activity can vary, and correspondingly the computations that can be carried out in the output-null space.”

A3) The paper is based on a center-out BCI task. However, there were no figures or data included on the behavioral performance. After reading the methods, I saw a paper referenced. It would be nice to include a figure on behavioral performance in the current manuscript, especially given that a third monkey appears to have been added following the previously published paper. It is helpful to the reader to have the behavioral data present in the current paper rather than having to go back to the previous work.

We have added a supplementary figure including a summary of behavior for the included experiments (Figure 1—figure supplement 1). We have also updated the Materials and methods to clarify that the experimental data is the same as in Golub et al. (2018), and that indeed we now include data from a third monkey that was not included in Sadtler et al. (2014) (see end of response to A1 above).

The following points require clarification or rewriting in the main text and/or addition to Discussion:B1) "Minimal intervention principle" (MIP, in this review) is used to describe a set of alternatives that – for us at least – do not really embody this notion. […] We are by no means rejecting hypotheses 3-5 as being irrelevant, but we think that MIP might in fact have more in common with hypothesis 6 ("fixed strategy") in that optimal control of a few activity directions is likely to cause correlated activity across potent- and null-directions due to the intrinsic dynamics of the circuit.

We agree with the reviewers that another version of the minimal intervention principle (MIP) may apply *upstream* of M1 (so that both the cursor and M1 are the plant), and that from this perspective, one might expect that this would induce coupling between the output-null and output-potent dimensions of the BCI, and thus be entirely consistent with our findings of a Fixed Distribution. This exact hypothesis would be difficult to test directly with our current data (because we are not recording from any upstream areas), but could in the future be investigated by recording simultaneously from areas likely to drive M1 (e.g., PMd).

As the reviewers point out, there are many ways of applying MIP to neural activity, and in our case we interpret the cursor as the plant and M1 as the input to the plant. The fact that MIP, as we implement it in our hypotheses, did not accurately predict the output-null distribution does not rule out the possibility of some other implementation of MIP being relevant to describing neural activity. In our Discussion paragraph on optimal feedback control (OFC), we have clarified that our current application of OFC/MIP does not rule out the possibility that other versions of OFC/MIP are relevant to predicting neural activity:

“Overall, our work does not rule out the possibility that OFC is appropriate for predicting neural activity. […] This could induce coupling between the output-potent and output-null dimensions of the M1 activity, and thereby yield predictions that are consistent with the findings presented here.”

In this work we refer to the minimal intervention principle as the idea that activity in task-relevant (output-potent) dimensions can be corrected independently of activity in task-irrelevant (output-null) dimensions (Todorov et al., 2002; Valero-Cuevas et al., 2009; Diedrichsen et al., 2010). Typically, this idea has been applied to muscle activity, whereas in the present study we apply MIP to M1 activity. We have clarified our definition of MIP for neural activity in the Results:

“An explanation of this variability asymmetry is the “minimal intervention” principle (Todorov et al., 2002; Valero-Cuevas et al., 2009; Diedrichsen et al., 2010), which states that while variability in output-potent dimensions should be corrected to ensure task success, variability in output-null dimensions can be left uncorrected because it does not lead to deficits in task performance. While this principle has been used to explain muscle activity, here we investigate whether it also explains neural activity.”

B2) Concerning the last hypothesis (fixed distribution), we believe it could be tested using any relevant distribution of activity, not only the one recorded under the "previous mapping"; it seems that all that matters is that you have a "reference distribution" that predicts some degree of coupling/correlations between the two sets of potent/null directions that define the current mapping. This, for example, could be the activity used to define the "intrinsic manifold". Have the authors considered testing this last hypothesis using such prior data? If it explains the structure of output-null activity equally well, this could add to the argument that output-null activity is mainly constrained by the dynamics of the circuit under control. The authors should comment on this, explain how this would fits into the results in the paper as a whole.

Indeed, the neural activity that is used to define the intrinsic manifold could be used as a “reference distribution” to predict the output-null activity expected during control of the first or second BCI mappings. However, this data is usually a small number of trials, and so would not provide a large enough sample of the neural activity to predict the output-null distribution expected during control of the BCI mapping. Having a sufficiently large number of samples is necessary because all but two of our hypotheses (Minimal Firing and Minimal Deviation) must use this activity to form predictions of the distributions of output-null activity expected for each direction of cursor movement.

In support of the idea that all you need is a “reference distribution” of neural activity, we predicted output-null activity in the opposite order to what was shown in the main text, i.e., we predicted output-null activity during the first (“Intuitive”) mapping using the activity observed during the second (“Perturbed”) mapping (Figure 5—figure supplement 2). The results were similar to those in the main text.

Related to this, it seems possible that the motor cortex activity could drive muscle movements, in addition to the BCI cursor. In this case, some of the output null dimensions could actually be related to output potent dimensions, just not for cursor movement. In this case, the so-called redundant or output null dimensions might not actually be redundant. Rather, they would be serving a different purpose, which could explain possible structure in their activity distributions. In this sense, I would not consider these redundant dimensions. Rather, the activity could be driving movements in a higher than 2D space if one considers both the BCI output and muscle movements. It would be helpful to see some description of how the authors have ruled out this possibility or why they have considered it to be a non-significant issue.

This is an interesting point to consider. In general, we agree with the reviewers that, if the dimensions responsible for moving the arm overlap with both the output-potent and output-null dimensions of the BCI, this might explain the coupling we observe between the output-potent and output-null dimensions. However, in these experiments, the animal’s arm was not moving during BCI control (see Extended Data Figure 5 in Sadtler et al., 2014). Thus, the activity we study here resides within the arm’s output-null dimensions. This implies that in our recordings the arm’s output-potent dimensions do not overlap with either the output-potent or the output-null dimensions of the BCI, and so this is unlikely to explain the coupling we observed between the output-potent and output-null dimensions of the BCI.

Overall, being unaware of extra output-potent dimensions would likely make the predictions of the Fixed Distribution hypothesis worse, not better. The reason for this is as follows. The Fixed Distribution hypothesis predicts that the distribution of activity in output-null dimensions depends upon the corresponding output-potent activity. Under this hypothesis, without knowing all of the output-potent activity, we cannot fully describe the corresponding output-null distribution. So if there is an output-potent dimension that we have not accounted for in our analyses (e.g., the dimension that helps keep the arm still), we would likely improve our predictions if we did account for this dimension. The fact that we predict so accurately (13% histogram error on average, with the lowest possible error being 7%) without knowing all the potent dimensions is then evidence that these extra output-potent dimensions, if they exist, would likely not provide substantial additional predictive power. We have updated the Discussion to address this possibility:

“It is interesting to consider the relationship between arm movements and BCI cursor movements (Orsborn et al., 2014; Vyas et al., 2018). […] The fact that we were able to accurately predict the output-null distributions (13% histogram error on average, with the lowest possible error being 7%) without knowing all the potent dimensions is then evidence that these extra potent dimensions, if they exist, would not provide substantial additional predictive power.”

B3) Are the authors confident that, following a switch in the decoder, the animal has enough time with the new decoder for any expected null space changes to occur? The trial count into the hundreds does seem substantial, so this is not a big concern, though the authors might wish to comment on how long the animal had with the new decoder (especially whether this was over one or multiple sessions/days), in relationship to timescales of plasticity and expected changes in activity, in the Discussion.

We agree that this is an important point to consider, which we bring out in Discussion

“Second, this study focused on short timescales, where we predicted output-null activity within one to two hours of subjects learning a new BCI mapping. […] Given repeated practice with the same BCI mapping across days and weeks (Ganguly et al., 2009), it is possible that there are different or fewer constraints on neural redundancy than what we found here.”

Performance with the second mapping was generally not as good as with the first mapping (Figure 1—figure supplement 1; see response to A3 above). To ensure that any potential results were not due to incomplete learning of the second mapping, we did two things. First, we restricted our analyses to the trials where monkeys had established stable performance with the new BCI mapping, as described in Materials and methods:

“We further selected those [sessions] in which the animal learned stable control of the second mapping. […] This was done to ensure that we were analyzing trials for which animals used a consistent strategy for selecting activity patterns.”

Second, we considered the possibility that animals may have incorrectly estimated which neural dimensions were output-potent in the BCI mapping and which were output-null. To control for this, we estimated which dimensions the animals believed to be output-null (Golub et al., 2015), and performed our analyses in the main text relative to these dimensions. We also repeated our analyses using the true output-null dimensions of the BCI and found similar results (Figure 5—figure supplement 3), suggesting that animals’ mis-estimates of the BCI mapping did not affect our analyses. We describe this approach in Results:

“Additionally, because animals learned to use the BCI mappings through trial and error, it is possible that the animals' assumptions about the output-null dimensions do not align perfectly with the actual output-null dimensions of the BCI mapping. […] The results in the main text are based on this internal model, and we show in supplemental figures that all results still hold when using the actual BCI mapping.”

B4) The BCI case is quite nice for studying this type of question. However, the dimensionality of the neural population controlling the BCI output is rather limited compared to perhaps a natural case of an entire motor cortical region capable of controlling a set of muscles. Dimensionality seems like it should play a significant role in considerations of redundancy, both on the neural end and the output end. It would be helpful to see more analysis or discussion of how neural and behavioral dimensionality, and perhaps most significantly the difference between the two, could play a role in interpretation and analysis of redundancy. For example, intuitively it seems possible that redundancy might be less constrained if 10,000 electrodes were used for BCI rather than 96. Is this true?

These are all great points. First, we address the impact of neural and output dimensionality on our results. In our setup, the dimensionality of the population activity is 10 (for the ~90 neural channels in our recordings), and the output (BCI) dimensionality is two. Given how many more dimensions of population activity there are than output dimensions, it came as a surprise to us that conditioning on only two neural dimensions (as we do in the present study) could provide so much explanatory power for predicting the distribution in the remaining neural dimensions. This suggests that many of the dimensions of population activity are coupled, i.e., changing the activity along some dimensions may also lead to changes along other dimensions. During arm movement control, output dimensionality and presumably the neural dimensionality are larger than in our setup. We speculate that during arm movements, many of the null dimensions will remain coupled with the potent dimensions, thereby yielding results similar to what we found in the present study. Of course, as the reviewers appreciate, the current barrier to testing these hypotheses using arm movements is the difficulty in identifying the output-potent dimensions. Until then, future work could examine the effects of larger output dimensionality on redundancy by repeating our analyses with a higher-dimensional BCI effector, such as a multiple degree of freedom robotic limb (e.g., Wodlinger et al., 2015). We have added the following to Discussion to address this point:

“Given how many dimensions of population activity there are (in this case, 10), it is somewhat surprising that conditioning on only the two output-potent dimensions could provide so much explanatory power for predicting the distribution in the remaining neural dimensions. […] Future work could examine whether animals can be trained to uncouple dimensions, as well as the effects of larger output-potent dimensionality on redundancy, by repeating our analyses with a higher-dimensional effector, such as a multiple degree-of-freedom robotic limb (e.g., Wodlinger et al., 2014).”

As for the impact of recording from 10,000 electrodes rather than 96, this is an interesting question. Because we analyzed neural activity in the low-d space identified using factor analysis (the 10D “intrinsic manifold” of Sadtler et al., 2014), we believe that our results would be qualitatively similar if we recorded from more units. The reason comes from Williamson et al. (2016), who used data from V1 recordings as well as network models to study how dimensionality and shared variance change as you record from more units. Using V1 data, they found that as the number of units recorded from increased, estimates of neural dimensionality also increased, but the same dominant (i.e. high covariance) modes that were identified with 20 units were also identified with 80 units. They then observed the same trend when using network simulations, where the number of simulated units varied from 20 units to 500 units. We repeated the same analyses using the recordings in the present study (varying the number of neural units from 2 to 85) and found similar results as the V1 data of the Williamson et al. study (Appendix—Figure 1). So even though recording from more neural units reveals more neural dimensions (Appendix—Figure 1A), these additional dimensions explain a very small proportion of the overall shared variance (Appendix—Figure 1 plateau in panel B). Thus, the effective dimensionality remained unchanged when going from 30 to 85 units. Furthermore, we found that these dimensions with high shared variance (i.e. the intrinsic manifold) were similar when increasing number of recorded units (Appendix—Figure 1C). Because all analyses in this work were carried out within the intrinsic manifold, we believe that our results would likely continue to hold with an even larger number of recorded units. We have added this analysis to Materials and methods, including a new Appendix—Figure 1:

“To understand how our results might change if we recorded from more neural units, we assessed the dimensionality and shared variance of population activity with a varying number of units (Williamson et al., 2016) (Appendix—Figure 1). […] Thus, recording from more units beyond the ~85 units that we recorded in these experiments is not likely to substantially change the results reported in this work.”

B5) The authors limit their analysis to the in-manifold perturbations of their experiment. The dataset has also out-of-manifold perturbations. Some comments on what is to be expected here seem worth mentioning in the Discussion.

For these analyses we need monkeys to show proficient cursor control using the two mappings presented in each session. However, as shown in Sadtler 2014, monkeys were less able to learn to control OMPs (outside-manifold perturbations) than WMPs (within-manifold perturbations) within a day. For OMPs, we would expect Fixed Distribution to do well for trivial reasons: If the monkey has not learned to control the cursor under the new mapping, the neural activity would likely not have changed much from what it was during the previous mapping. Thus, we decided to focus on WMPs in the present work.

We have clarified this point in the Materials and methods:

“The data analyzed in this study was part of a larger study involving learning two different types of BCI mapping changes: within-manifold perturbations (WMP) and outside-manifold perturbations (OMP) (Sadtler et al., 2014). […] Among the WMP sessions, we further selected those in which the animal learned stable control of the second mapping (42 selected and 12 discarded).”